# Substantial aircraft contrail formation at low soot emission levels

Christiane Voigt[1,2 ✉], Raphael Märkl[1,2], Daniel Sauer[1], Rebecca Dischl[1,2], Charles Renard[3], Katharina Seeliger[3], Fangqun Yu[4], Stefan Kaufmann[1], Tiziana Bräuer[1], Tina Jurkat-Witschas[1], Gauthier Le Chenadec[3], Julien Moreau[3], Emiliano Requena-Esteban[3], Nicolas Bonne[5], Margaux Vals[5], Amandine Roche[6], Joseph Zelina[7], Andreas Dörnbrack[1], Lisa Eirenschmalz[1], Christopher Heckl[1], Elisabeth Horst[1], Michael Lichtenstern[1], Andreas Marsing[1], Gregor Neumann[1], Anke Roiger[1], Monika Scheibe[1], Paul Stock[1], Andreas Giez[1], Georg Eckel[1] & Patrick Le Clercq[1]

Contrail cirrus clouds are a main contributor to the climate forcing from aviation[1]. Yet, the number of contrail ice crystals forming behind aircraft with modern lean-burn engines is unknown. Theory spans a four orders of magnitude range in ice crystal numbers[2,3]—rendering related climate effects unpredictable. Here we show that lean-burn combustion reduces soot particle number emissions by three orders of magnitude compared with conventional rich–quench–lean engines[4,5]—but does not significantly decrease volatile particles or contrail ice crystal numbers—both can exceed $10^{15}$ particles per kg of burned fuel. Our findings arise from in-flight observations behind an A321neo aircraft with lean-burn engines, thus providing real-world confirmation of some laboratory work[6] and narrowing the range of theoretical expectations. Our results indicate that the tested lean-burn engine configurations alone are unlikely to reduce the warming effect of contrails, suggesting that modifications of fuel composition and lubrication oil venting architecture may be required. We show that contrail ice particle numbers in the low-soot regime can be reduced by using low-sulfur fuels and that organic fuel constituents and lubrication oil vapours can increase contrail ice particle numbers. Future research should explore how reductions in volatile particles, apart from soot, affect contrail ice formation.

Aviation plays a vital part for mankind, industry and economy and the transport of goods and people. Aircraft also contribute to climate change mainly by carbon dioxide ($CO_2$) emissions and by formation of contrail cirrus. Notably, the annual mean effective radiative forcing from contrails is almost on par with that from the carbon dioxide emissions of aviation since the historical start of air traffic[1,7]. Meanwhile, global air traffic has recovered from the 2020 pandemic[8] and is expected to increase by a factor of 2–3 by 2050 (ref. 9). Hence, there is an urgent need for an international aviation strategy that reflects the essential role of aviation for economy and global competitiveness— and that also curbs aircraft emissions, contrails and related climate effects[10]. This is also expressed in environmental efforts by the International Civil Aviation Organization in their commitment to fly net-zero carbon emissions by 2050, signed by 16 main players in the aviation sector[11]. Regulators have reacted to this aviation challenge, and the European Union has released the Destination 2050 Roadmap[12,13], which sets limits on aircraft $CO_2$ emissions and may demand the monitoring and reporting of the non-$CO_2$ effects of aviation[14]. Parallel to a debate on uncertainty or the best-suited metric[1], academia and industry have made considerable progress to better understand the non-$CO_2$ effects

of aviation. Recent studies[15,16] developed probability distributions of aircraft $CO_2$ and non-$CO_2$ climate impacts to assess the risk of mitigation measures with opposing effects on climate. The analysis in these studies favours the reduction of aircraft non-$CO_2$ effects for measures with $CO_2$:non-$CO_2$ trade off ratios smaller than 1:5. Unlike well-mixed $CO_2$ emissions with atmospheric lifetimes of many decades[17], contrail cirrus may persist at cold and humid cruise conditions for only several hours[18–20]. Hence, in contrast to $CO_2$, measures to reduce warming contrail cirrus would have an immediate effect on the climate, which is one of the levers required to meet the international climate targets.

Current developments of advanced solutions to reduce the total climate effect from aviation include alternative bio-based or synthetic fuels[4,21], engine and aircraft technology[9] as well as operational measures[22]. The latter includes avoiding the formation of contrails by flying above or below ice-supersaturated regions[23], in which warming contrails would form. However, these operational measures may come at the cost of slightly increased fuel consumption[22].

Another promising strategy to reduce the contrail climate effect is bio-based or synthetic aviation fuel (SAF)[24] produced with renewable energies, which can also have a reduced $CO_2$ footprint compared with

[1]Deutsches Zentrum für Luft- und Raumfahrt (DLR), Oberpfaffenhofen, Germany. [2]Johannes Gutenberg-Universität, Mainz, Germany. [3]Airbus Operations SAS, Toulouse, France. [4]Atmospheric Sciences Research Center, University at Albany, Albany, NY, USA. [5]Office National d'Etudes et de Recherches Aérospatiales (ONERA), Palaiseau, France. [6]Safran Aircraft Engines, Villaroche, France. [7]GE Aerospace, Cincinnati, OH, USA. ✉e-mail: Christiane.Voigt@dlr.de

**Table 1 | Composition of probed fuels and world average Jet A-1**

| Fuel | Fuel components | Hydrogen content (%m/m) ASTM D3701 | Carbon content (%m/m) calculated from H content | Sulfur content (ppmm) ASTM D5453 | Aromatic content (%v/v) ASTM D6379 | Naphthalene content (%v/v) ASTM D1840 |
|---|---|---|---|---|---|---|
| Jet A-1 | Conventional Jet A-1 | 14.1 | 85.9 | 195 (192) | 12.8 | 0.6 |
| Jet A-1* world average | World average Jet A-1 | 13.9 | 85.1 | 460 | 19.2 | 1.2 |
| HEFA-SPK | 100% HEFA- SPK | 15.3 | 84.7 | 3 | <1 | <0.1 |
| HEFA-blend | 64% HEFA+36% Jet A-1 | 14.8 | 85.2 | 75 | 4.7 | 0.3 |
| SPK-L | HEFA-SPK+low aromatics | 14.8 | 85.2 | 2 (9) | 8.4 | <0.1 |
| SPK-H | HEFA-SPK+high aromatics | 14.3 | 85.7 | <1 (4) | 17.6 | <0.1 |

Fuel composition, measurement method and properties of fuels used during NEOFUELS/VOLCAN; and world average Jet A-1 composition from ref. 49. Units are given as per cent by mass (%m), parts per million by mass (ppmm) or per cent by volume (%v) to account for fuel unit conventions. The hydrogen content was measured by ASTM D3701 with a repeatability of 0.09 and a reproducibility of 0.11. The sulfur content was measured by ASTM D5453, with a detection limit of 1 ppmm. With respect to uncertainties, 1 (5) ppmm sulfur was detected with a repeatability (r) of 0.2 (0.6) and reproducibility (R) of 0.6 (1.9), respectively. For SPK-H and SPK-L, the FSC was measured for all probes from the refuelling truck. Furthermore, the FSC measured in fuel samples from the aircraft after the flights are given in parentheses. Fuel samples of Jet A-1 shown in Fig. 2 and in Extended Data Figs. 2 and 3 may vary slightly and might contain small amounts of HEFA fuel, resulting in 192 ppmm sulfur in Jet A-1 for specific flights. The chemical properties of this blend are very close to pure Jet A-1, and it is referred to as Jet A-1 throughout this study.

conventional Jet A-1 fuel. Owing to their lower aromatic fuel content, SAFs lead to a reduction in soot particle emissions[4,5,24]. For conventional rich–quench–lean (RQL) engine technologies, the soot particles around 30 nm in size serve as one of the primary nuclei sources for contrail ice crystals[25], and a substantial reduction in soot and contrail ice crystals has been observed when burning low-aromatic SAFs[21,26,27]. This can reduce the lifetime of contrails[20] and their radiative forcing[7].

Although the effects of SAF on contrails and climate have been investigated on engines fitted with RQL combustor technologies, in-flight emissions and contrail data from modern lean-burn combustors are missing, which leads to uncertainties in current contrail climate impact predictions. Lean-burn engines are designed to improve engine emission performance through a fuel injection system and an airflow distribution that expands regions with low fuel-to-air ratios in the combustor[28], with the objective of lowering nitrogen oxides and soot particle emissions. Emission certification tests with lean-burn engines also indicate very low soot mass and number emissions at higher power settings[29].

Models[30,31] predict a near-linear relationship between emitted soot particles and contrail ice crystal numbers for current RQL engines. Moreover, a theoretical analysis[32] of flight data[21] from rich-burning engines suggests that fuels with a low sulfur content can lead to reduced contrail ice crystal activation. Conversely, recent flight measurements indicate that, for fuels with elevated sulfur content, ice crystal number concentrations may exceed soot particle emissions, pointing to a sulfur-induced modulation of the soot–ice correlation in the high-soot regime[33].

By contrast, in the low-soot regime ($\ll 10^{14}$ soot particles emitted per kg of fuel burned) anticipated for lean-burn engines, microphysical contrail models[3,31,34] predict a much wider range of possible ice crystal numbers, spanning four orders of magnitude, but these predictions lack experimental validation. In particular, there are no published particle number emission or contrail data for lean-burn engines at cruise, creating a substantial gap in data required to calculate and assess the related climate effects. Therefore, most climate models[7,35] assume very low initial ice crystal numbers for lean-burn aircraft—potentially underestimating their contrail-related climate effects. The data gap also affects future fleet simulations[9], potentially resulting in unreliable predictions, particularly for emerging technologies such as hydrogen-powered aircraft[36].

## Flight strategy and measurement concept

Here, we provide the first in-flight dataset on engine emissions and contrail properties produced by an aircraft with lean-burn engines. During the NEOFUELS/VOLCAN (VOL avec Carburants Alternatif Nouveaux)

campaign, the Deutsches Zentrum für Luft- und Raumfahrt (DLR) partnered with the aircraft manufacturer Airbus and the engine manufacturer SAFRAN/CFM to measure the emissions of an A321neo equipped with state-of-the-art LEAP-1A engines. Under normal operation, the combustor of the LEAP-1A engine is staged to the lean-burn mode at high power settings during take-off, climb and cruise phases. There, the inner pilot and the outer main fuel injectors are active, while during taxi, descent and approach phases at low power, the outer injector is switched off to guarantee stable combustion in a kind of rich-burn mode. Engine control adjustments were developed by the engine manufacturer specifically for this campaign to overwrite the normal fuel injector control law in the automated engine control (FADEC) and allow for defined operation either in controlled lean-burn or in forced rich-burn combustion mode at the same combustor inlet temperature.

The fuel system of the A321neo has three separate tanks, which can be switched to the different engines independently and enable testing of different fuels during the same flight. The A321neo was fuelled with conventional petroleum-based fossil kerosene (Jet A-1) as well as 100% bio-based hydrotreated ester and fatty acids synthetic paraffinic kerosene (HEFA-SPK) with different hydrogen, aromatic, naphthalene and sulfur compositions (Table 1), both provided by Total Energies. To enable the investigation of specific fuel component effects on contrails, the low-sulfur HEFA-SPK was also blended with fossil-based mono-aromatics. An aromatic content of 8% was targeted for the SPK-L, which is the lower limit of the ASTM D7566 standard for the specification of fossil fuel and sustainable aviation fuel blends[37]. Aromatics were added to the second SPK-H blend to reach 17.6% to meet the lower density limit of the regulation. The aircraft and the engines were specifically cleared for operations with 100% HEFA-SPK. The composition and properties of the investigated fuels are given in Table 1.

The DLR research aircraft Falcon 20E[38] was equipped with a comprehensive set of instruments to measure trace gases, in particular, $CO_2$, water vapour and nitrogen oxides, as well as properties of aerosol and contrail ice particles, and meteorological data, as described in detail in the Methods. The Falcon chased the A321neo at close distances of 40–250 m equivalent to 0.2–1.4 s plume age during repeated emission measurements in different engine modes and for different fuels (Extended Data Fig. 1). In one flight, the left engine was also probed while burning Jet A-1, whereas in all other flights, the right engine was probed. As contrail ice crystal formation can be incomplete at less than 1 s plume age, contrails were probed in the far field at distances of 6–29 km from the preceding aircraft in traffic-restricted airspaces over the Atlantic and the Mediterranean. After the flights, the particle emission and contrail data were dilution-corrected and correlated to the fuel consumption and the combustion mode. Detailed information on the aircraft, engines, instrumentation, data evaluation and models is provided in the Methods.

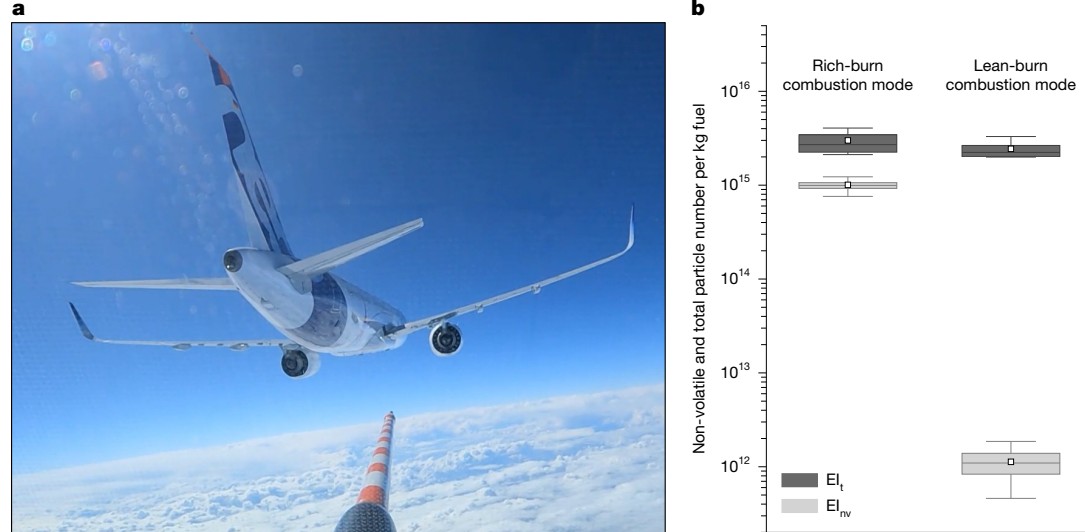

**Fig. 1 | Flight measurements show a three orders of magnitude reduction in non-volatile soot particle emissions in lean-burn compared with forced rich-burn combustor configuration. a**, Noseboom of the research aircraft Falcon of DLR chasing the Airbus A321neo equipped with CFM LEAP-1A lean-burn engines during an emission flight on 7 March 2023, probing the 0.2–1.4 s-old exhaust at distances of 40–250 m. **b**, Emission indices of total particles ($EI_t$, $d > 5$ nm, dark grey) and non-volatile particles ($EI_{nv}$, $d > 14$ nm, light grey) emitted per kg fuel in forced rich-burn and lean-burn engine conditions for reference Jet A-1 fuel. Results show means, medians, 25 and 75 percentiles, minimum and maximum (see also Extended Data Table 1). Photo in **a** is reproduced with permission from DLR, CC BY-NC-ND 4.0.

## Particle emissions by lean-burn engines

The particle number emission index ($EI_x$) measures the number of non-volatile or total particles emitted per kilogram of burned fuel, assuming that the fuel carbon content is completely converted to $CO_2$ (Methods). Here, we assign the non-volatile particle number emission index $EI_{nv}$ to soot particles, because soot particles are the main emission component in the aircraft exhaust in the size range larger than 14 nm. We measured a median non-volatile particle number emission index $EI_{nv}$ in the forced rich-burn mode of the LEAP-1A engine on the A321neo of $1.0 \times 10^{15}$ (range $0.8–1.1 \times 10^{15}$) particles per kg fuel burned for the Jet A-1 fuel probed during the campaign (Fig. 1 and Extended Data Table 1). Although forcing the combustor to operate in rich-burn mode does not reflect the typical and optimal combustor setting at cruise, it is used here to compare rich-burn and lean-burn engine emissions at similar power settings in terms of combustor inlet temperature T30. The measured $EI_{nv}$ is lower compared with Jet A-1 data from older engines of the IAE V2500 engine series with higher soot emissions[21] ($5.0 \times 10^{15}$ kg$^{-1}$ fuel). Soot particle emissions are in the range of cruise data from a modern Rolls-Royce Trent-XWB-84 engine[5] ($1.0 \times 10^{15}$ kg$^{-1}$ fuel) and higher than emissions from an older CFM56-2-C1 engine[4] (about $3.0 \times 10^{14}$ kg$^{-1}$ fuel) measured at different ambient and engine conditions.

Particle number emission indices were also measured in the lean-burn combustion mode of the LEAP-1A engine on the A321neo. Here, the $EI_{nv}$ is massively reduced by three orders of magnitude to a median of $1.0 \times 10^{12}$ (range $0.5–1.9 \times 10^{12}$) kg$^{-1}$ fuel. The $EI_{nv}$ is slightly above the detection limit determined by the ambient aerosol background concentration plus three times its standard deviation. The reduction in soot particle numbers of the LEAP-1A engine in typical lean-burn cruise conditions is large. As soot particles are the preferred nuclei for contrail ice crystals[3,25], the question arises whether ice crystal numbers in contrails are reduced by the same order of magnitude.

## Contrail ice crystal numbers in lean-burn mode

To this end, contrail measurements were conducted with both engines operating in the typical lean-burn cruise mode. Persistent contrails were detected 6–29 km downstream of the preceding A321neo in ice-supersaturated conditions. The apparent contrail ice particle number emission index ($EI_{ice}$) is calculated analogously to $EI_{nv}$. For the contrails formed from Jet A-1 fuel under lean-burn conditions, we measured a mean $EI_{ice}$ of $1.6$ ($\pm 0.3$) $\times 10^{15}$ kg$^{-1}$ fuel (Fig. 2), approximately a factor of 1,000 higher than the corresponding soot particle emission index. These measurements represent the first contrail observations downstream of engines operating in lean-burn combustion mode. The contrail ice crystal numbers exceed soot particle emissions by three orders of magnitude; thus, measured (>14 nm) soot particles alone cannot explain the observed contrail ice particle numbers.

To explore other potential ice nuclei, we direct the attention to the total particle number emission index $EI_t$, including non-volatile and volatile particles with diameters larger than 5 nm, measured in the near-field in lean-burn conditions (Fig. 1). For Jet A-1, $EI_t$ has a median value of $2.1 \times 10^{15}$ kg$^{-1}$ fuel (range $2.0–3.3 \times 10^{15}$ kg$^{-1}$ fuel) promoting the question whether those volatile particles could play a role in contrail ice formation at very low soot emission levels.

## Contrail formation on volatile particles

Contrail formation theory[2,3] suggests that fuel sulfur species may be oxidized to sulfuric acid, which may be scavenged by co-emitted soot particles in the rich-burn combustion mode. At low soot emission levels, this condensational sink is reduced, and fuel sulfur components may nucleate to sulfate aerosol particles[3]. To investigate whether these tiny volatile particles may influence contrail formation, we use contrail measurements from two fuels with different sulfur contents. Also, to avoid ambiguities from varying ambient conditions, we constrain the contrail data set to measurements at similar temperatures (218 K, 105–112% relative humidity with respect to ice ($RH_i$); Extended Data Table 2). A statistical analysis[39] showed the best comparability of Jet A-1 and HEFA-SPK contrail data for a flight, in which the HEFA-SPK was contaminated by an overnight leakage between the tanks, which resulted in a HEFA-blend (Table 1). The measured mean $EI_{ice}$ of $1.6$ ($\pm 0.3$) $\times 10^{15}$ kg$^{-1}$ fuel for Jet A-1 with 192 ppmm sulfur content can be compared with an $EI_{ice}$ of $0.5$ ($\pm 0.2$) $\times 10^{15}$ kg$^{-1}$ fuel for HEFA-blend with 75 ppmm sulfur. Contrail ice crystal numbers are reduced by approximately a factor of 3 for the lower-sulfur HEFA-blend, providing an initial indication

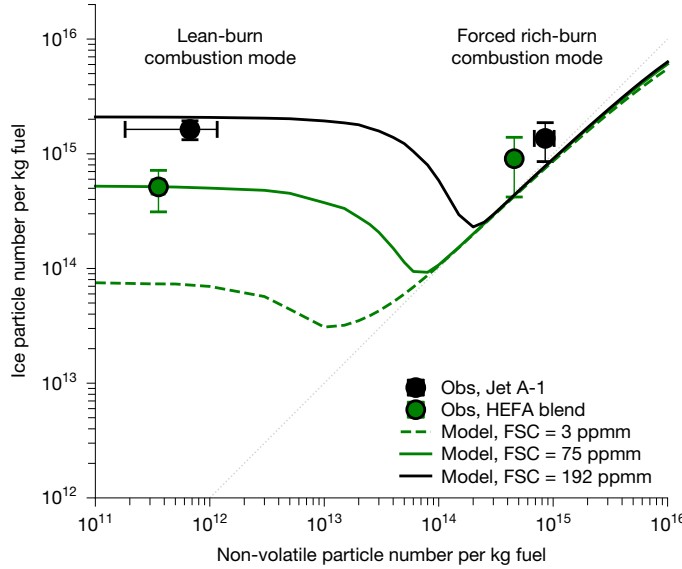

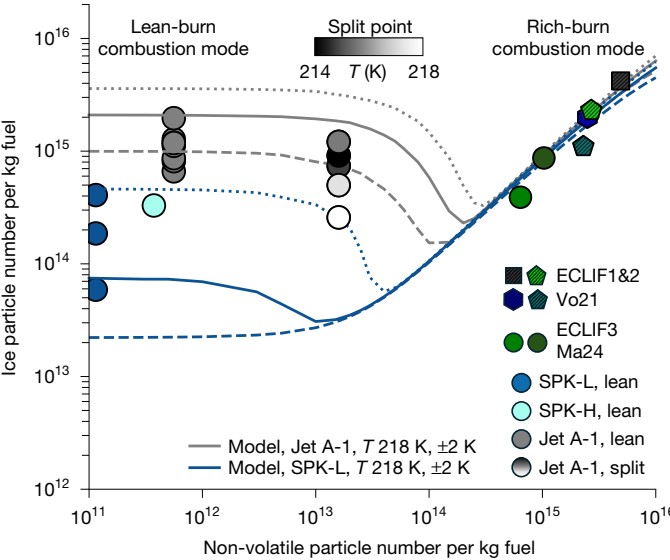

**Fig. 2 | Theory constrained by emission and contrail observations can explain contrail ice formation on volatile particles under low-soot, lean-burn combustor conditions.** Dependence of contrail ice crystal numbers on combustion mode and fuel composition. The symbols show mean and arithmetic standard deviation of non-volatile ($EI_{nv}$, $d > 14$ nm) and ice ($EI_{ice}$, $d > 0.6$ μm) particle number emissions indices measured under lean-burn and forced rich-burn combustor configurations for Jet A-1 (black) and a HEFA-blend (green) at 218 K ambient temperature. The lines denote simulations with the updated ACM model[3] at comparable conditions (218 K, 10 K below $T_{SA}$; refs. 39,50) for lean- and rich-burn combustion modes and three fuel composition cases (Jet A-1 with 192 ppmm sulfur; HEFA-blend with 75 ppmm sulfur; and HEFA-SPK with 3 ppmm sulfur; see also Table 1). The grey dotted line indicates the 1:1 relationship. More details on measurement conditions for the near-field emission and the far-field contrail measurements are provided in the Methods and Extended Data Tables 2 and 3. Reduced ice particle number emission indices were measured and modelled for low-sulfur low-aromatic HEFA-blend compared with Jet A-1.

**Fig. 3 | Reduced contrail ice crystal numbers were measured and modelled for low-aromatic, low-sulfur fuels in all combustion modes and for contrails formed at higher ambient temperatures.** Dependence of contrail ice crystal numbers on combustion mode, fuel composition and temperature. The symbols show non-volatile ($EI_{nv}$, $d > 14$ nm), and ice ($EI_{ice}$, $d > 0.6$ μm) particle number emissions indices per kg fuel, binned in 1 K $\Delta T_{SA}$ temperature intervals[39,50] for Jet A-1 (grey circles), as well as low- and high-aromatic HEFA-SPK blends (blue and cyan circles) in lean-burn and an intermediate split point between lean-burn and forced rich-burn combustor condition. The ambient temperatures are colour-coded for the split-point Jet A-1, with darker colours indicating colder temperatures. Data from previous campaigns with RQL engines are included for reference to denote the soot-rich combustor configuration (ECLIF1&2: Vo21 (ref. 21); ECLIF3: Ma24 (ref. 27)). The lines show results from the updated ACM contrail model[3] for different ambient temperatures (216 K, 218 K and 220 K) for Jet A-1 (grey) and low-aromatic low-sulfur SPK-L blend (blue), showing the combined effects of fuel composition and temperature on contrail ice crystal numbers. More details on conditions for the near-field emission as well as the far-field contrail measurements are provided in the Methods and Extended Data Tables 2–4.

that the reduced fuel sulfur content may contribute to the observed decrease in initial ice crystal numbers for the HEFA-blend.

Sulfur species may not be the only components forming new volatile particles in the engine exhaust. Recent laboratory, ground and model studies suggest a potential effect from organic and lubrication oil vapours on volatile particle formation[6,40–43]. As composition measurements of small few nm-sized volatile particles do not exist today because of instrumental limitations, we use a model to investigate contrail formation pathways in the low-soot regime.

## Contrail formation theory

Apart from contrail ice nucleation on soot and volatile sulfate aerosol particles, the aerosol and contrail microphysics (ACM) model[3] has been further developed for this study to explicitly simulate the nucleation and the condensation of lubrication oil vapours vented into the engine exhaust. Also, the model now considers the nucleation of low-volatile organic aerosol from incomplete combustion of fuel hydrocarbon components[41]. Details on the new model setup are given in the Methods. The simulation results in the low-soot regime show a good agreement with contrail measurements at similar ambient conditions (Fig. 2). For sulfur-rich Jet A-1 fuel, the model suggests that newly formed volatile sulfate aerosol, with minor contributions from organic and lubrication oil particles, may explain measured contrail ice crystal numbers in the low-soot regime. The results also show a strong dependence of contrail ice crystal numbers in the lean-burn regime on the fuel sulfur content. Sulfur-rich fuels tend to produce

higher numbers of sulfate aerosol particles, which can act as ice nuclei at low soot emission levels.

Ice crystal numbers were reduced by approximately a factor of 3 for the lower sulfur HEFA-blend, but remained substantial. For this fuel, volatile organic species from lubrication oil vapours vented into the hot centre core of the engine exhaust, together with unburned fuel hydrocarbon vapours, may contribute to the observed volatile particle formation and the associated ice crystal numbers. The ACM model reproduces the observations when using lubrication oil and organic vapour emission indices of 120 and 50–66 mg kg$^{-1}$ fuel, respectively, assuming that only a fraction of those vapours forms new particles[41]. The model also assumes that 3% of the fuel sulfur content is converted to sulfate aerosol in the early exhaust plume (Methods). For ultralow-sulfur fuels with 3 ppmm sulfur, the model predicts an order of magnitude reduction in contrail ice particle numbers, consistent with observations for the SPK-L fuel in the lean-burn mode (Fig. 3). Under these ultralow-sulfur conditions, the fuel sulfur contribution to volatile particle formation may be negligible, and organic vapours and lubrication oil emissions potentially dominate volatile aerosol nucleation and growth and subsequent contrail ice nucleation.

A sensitivity study with the microphysical contrail model in ref. 34 also indicates an effect of changes in soluble organics on contrail formation in the low-soot regime (Extended Data Fig. 3). Apart from the classical contrail ice nucleation pathway on soot[30–32], the good agreement between observations and model results for the four contrail

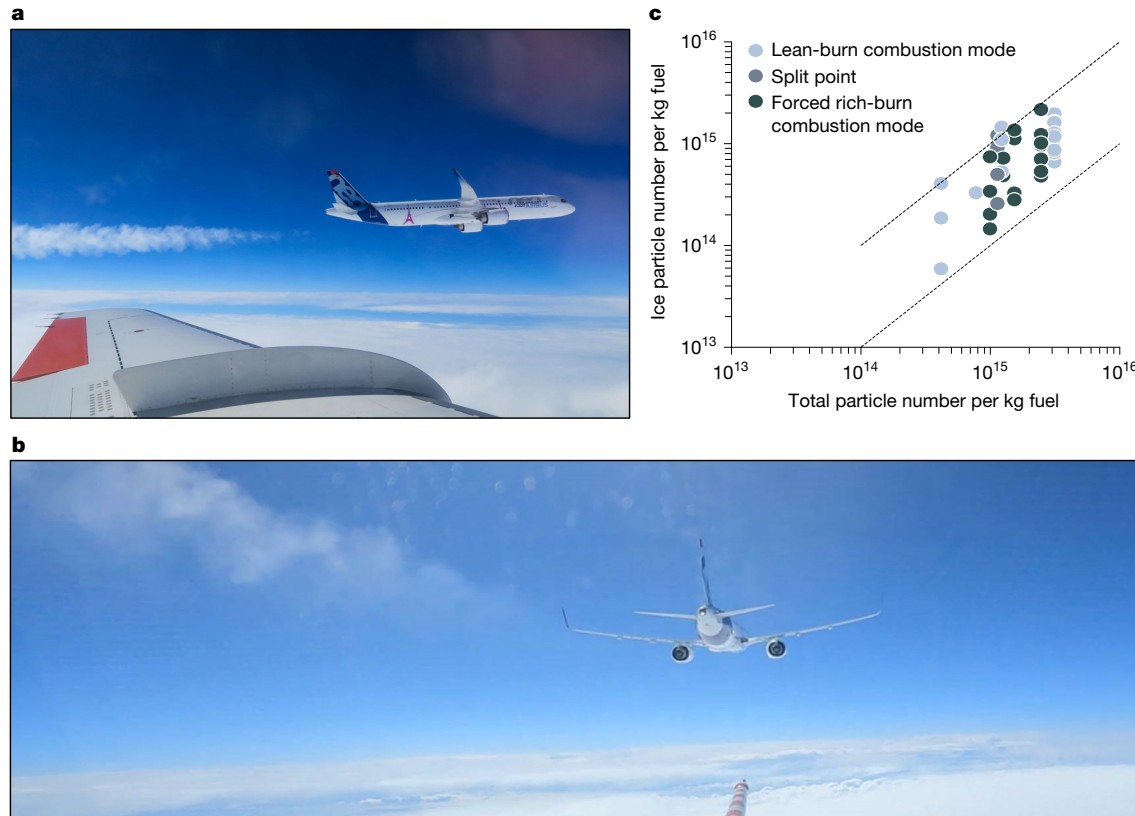

**Fig. 4 | Contrail ice particle numbers decrease with decreasing total particle emissions (non-volatile and volatile), independent of the combustion mode. a**, Photo of a contrail forming behind the A321neo taken from the research aircraft Falcon. **b**, Photo of an A321neo contrail in lean-burn combustion mode at temperatures close to the contrail formation threshold burning the same fuel in both engines. The larger contrail optical thickness of the left compared with the right engine might point to differences in volatile particle emissions by the two engines. **c**, Correlation between measured total (that is, volatile and non-volatile; $EI_t$, $d > 5$ nm) and ice ($EI_{ice}$, $d > 0.6$ μm) particle emission indices for all fuels and engine operating modes binned in 1 K $\Delta T_{SA}$ temperature intervals $T_{SA}$ (ref. 39). The dashed lines bounding the observations indicate the 1:1 and the 1:10 relationships. Further details on conditions for emission and the contrail measurements are provided in Extended Data Tables 2–4. Photos in **a** and **b** are reproduced with permission from DLR, CC BY-NC-ND 4.0.

cases under comparable ambient conditions (Fig. 2) confirms an additional contrail ice nucleation pathway—potentially from newly formed volatile sulfate, lubrication oil and organic particles, followed by condensational growth of the low volatile vapours onto the previously formed aerosols.

Ice particle size distributions (PSDs) from the contrail observations under comparable ambient conditions are shown in Extended Data Fig. 2. We measured slightly larger initial ice crystal sizes in contrails from HEFA-blend compared with Jet A-1 in lean-burn and forced rich-burn engine conditions. In the HEFA-blend cases, engine-emitted water vapour may be distributed evenly among a lower number of contrail ice crystals, resulting in increased ice crystal sizes, as observed previously in ref. 21 for semi-synthetic jet fuels.

## Temperature effect on contrail formation

An effect of ambient temperatures and water vapour mixing ratios on contrail ice crystal numbers has been observed previously[3,25–27]. To investigate the effect of ambient conditions on contrail formation, temperature-binned $EI_{ice}$ are compared with model results for varying ambient temperatures (Fig. 3). For Jet A-1, measured contrail ice crystal numbers generally increase with decreasing ambient temperatures. At cold temperatures, the maximum water supersaturation ratio in the plume is higher, which enables the activation of more small volatile particles and their nucleation into ice crystals. The model largely captures the range of observed $EI_{ice}$ for Jet A-1 and the HEFA-SPK blends with high and low aromatic content in forced rich-burn and lean-burn

regimes, potentially suggesting a temperature-dependent activation of the volatile organo-sulfate particles. The lowest contrail ice crystal numbers were observed for the SPK-L blend with 2 ppmm fuel sulfur content. For fuels with ultralow sulfur contents, the measured contrail ice crystal numbers in the low-soot regime can only be explained by ice nucleation on lubrication oil particles and volatile organic particles from incomplete fuel combustion. Although the model generally captures the temperature dependence in observed contrail ice crystal numbers, its ability to reproduce their absolute levels is limited, suggesting that additional factors can contribute to contrail ice formation.

Figure 3 also shows $EI_{nv}$ and $EI_{ice}$ from previous measurements behind RQL engines of the IAE-V2500 series[21], and a newer Rolls-Royce Trent-XWB-84 engine[27], confirming the model results in the rich-burn mode. A minimum in contrail ice crystal numbers is simulated between $10^{13}$ and $3 \times 10^{14}$ $EI_{nv}$, extending to lower values compared with previous simulations[2,3]. The minimum in contrail ice crystal numbers can depend on ambient temperature, fuel composition, combustion mode and engine type[3] and might represent a climate-optimal spot for contrail formation conditions by current RQL engines.

Our findings indicate that soot particles largely regulate the number of contrail ice crystals in the high-soot regime ($>10^{14}$ soot particles kg$^{-1}$ fuel), consistent with classical contrail formation theory[2,3]. Also, contrail ice crystal numbers may be modified by the fuel sulfur content, even for rich-burning engines, as suggested by previous in-flight measurements[21,27] and modelling studies[32,33]. In the low-soot regime, additional ice formation processes appear necessary—with evidence pointing to contrail ice nucleation on volatile sulfate aerosols

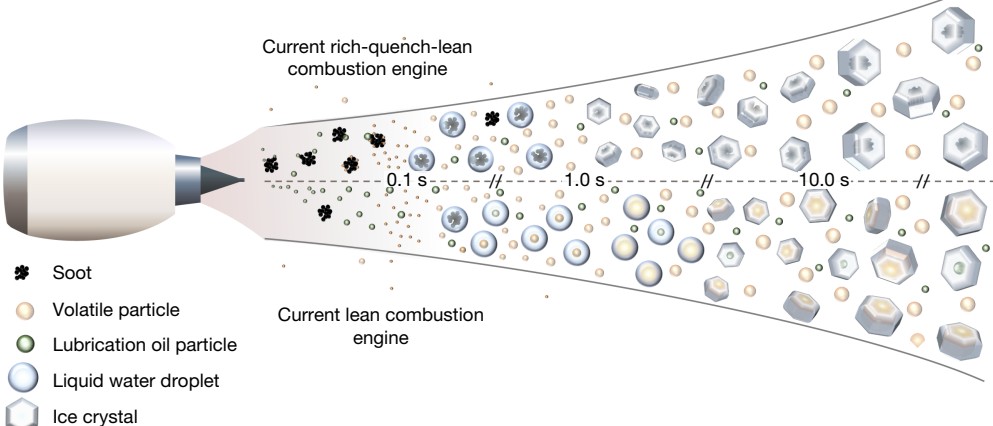

Current rich-quench-lean
combustion engine

0.1 s — // — 1.0 s — // — 10.0 s — // —

✳ Soot

○ Volatile particle

○ Lubrication oil particle

○ Liquid water droplet

⬡ Ice crystal

Current lean combustion
engine

**Fig. 5 | Schematic of ice formation processes for the RQL and lean-burn engine configurations.** Schematic of contrail formation for most current RQL (top) and lean-burn (bottom) combustor configurations. RQL engines emit large numbers of about 30 nm-sized soot particles, which are preferentially activated into water droplets on cooling, followed by homogeneous freezing and ice crystal growth. The emitted soot particles provide a condensational sink for the co-emitted sulfur, organic and lubrication oil vapours. Lean-burn engines emit substantially fewer soot particles; here, sulfur and organic vapours from fuel components as well as lubrication oil vapours vented into the hot core flow can form new volatile particles, or condense onto existing particles. On further cooling, ice freezes homogeneously in the volatile particles, followed by ice crystal growth mainly from uptake of ambient water vapour. Ambient aerosol may play a minor role in contrail ice formation for existing RQL and lean-burn combustor configurations. Visualization of the graphic was done by R.M., DLR.

for fuels with high and medium sulfur contents, and on volatile aerosols from lubrication oil and organic vapours for ultralow-sulfur fuels. More evidence for contrail ice formation on total (that is, volatile and soot) aerosol is found when plotting ice crystal numbers and total particles for rich and lean-burn combustor conditions and all fuel types (Fig. 4). The emission index $EI_t$ of total aerosol larger than 5 nm is in the range of $0.4–3.1 × 10^{15}$ kg$^{-1}$ fuel for all engine modes and the different fuels probed. Also, $EI_t$ is almost linearly correlated with $EI_{ice}$, and encompassed by the 1:1 and 1:10 line. The lowest $EI_t$ of $4 × 10^{14}$ kg$^{-1}$ suggests that ice nucleation on ambient background aerosol plays a minor role for the tested engine at the ambient conditions at which the measurements were taken. Generally, total aerosols, that is, volatiles and soot, contribute to contrail ice formation in both engine modes. In the rich-burn regime, $EI_{ice}$ are regulated by soot particle emissions, modulated by the fuel sulfur content, whereas in the lean-burn regime, $EI_{ice}$ are dominated by newly formed volatile organo-sulfate aerosols and lubrication oil particles, potentially facilitated by the condensation of organic vapours. The intermediate regime depends on many parameters and includes the point with the minimum contrail ice crystal numbers, which still needs to be explored.

## Understanding contrail formation

Potential ice crystal formation processes for RQL and lean-burn engines are shown in Fig. 5. In the high-soot regime of current RQL engines, ice preferentially nucleates on the larger soot particles. The emitted 30-nm-sized soot particles can be activated to form liquid droplets that freeze quickly to ice at cold temperatures. The nm-sized volatile particles formed by sulfuric acid, lubrication oil and unburned hydrocarbon vapours compete with larger soot particles for ice activation in the high-soot regime and may modulate contrail ice formation. In the almost complete absence of soot particles in the lean-burn regime (Fig. 5, bottom), ice nucleates on newly formed volatile aerosols that are activated into droplets. Ions emitted by the engines can activate low-volatility gaseous sulfuric acid and organic species to nucleate to volatile organo-sulfate aerosols[3,32,43], which provide sites for ice nucleation. Lubrication oil vapours[40–42] can condense on pre-existing aerosol or form new particles, which then can act as ice nuclei[6]. This results in similar ranges of contrail ice crystal number concentrations for aircraft with lean-burn or RQL engines, respectively. Thus, contrail ice numbers can be modified by changes in the fuel composition (sulfur, hydrogen, aromatic and naphthalene content) and might be affected by the release of lubrication oils through the oil venting system. Further experiments are required to better constrain the dependence of aerosol and contrail formation on fuel composition for specific engines and to further investigate the potential role of oil venting.

## Impact on fuel specification, engines and models

Our in-flight observations behind an aircraft with lean-burn engines show substantial volatile aerosol and contrail ice particle formation in the low-soot regime. Current ICAO CAEP standards[29] regulate direct engine emissions resulting from fuel combustion (mass and number of non-volatile particles, mass of NO$_x$, CO and unburned hydrocarbons). Our findings indicate that minimizing volatile particles may help to reduce initial contrail ice crystal numbers, representing a potential pathway towards clean aviation. However, more experimental data are needed to assess the associated impacts on microphysical and optical contrail properties over the entire contrail lifetime. The formation of a few very large oil droplets has been reported from ground emission measurements behind RQL engines with an oil venting system in the colder bypass flow[44–47]. By contrast, the venting of lubrication oil into hot areas of the core exhaust can lead to the formation of oil vapours[6,40], which then recondense on existing aerosol or nucleate new particles. Both processes can enhance contrail ice particle concentrations. Further experiments with targeted instrumentations on the ground[45–47] and in-flight are needed to explore whether nucleation of lubrication oil and organic vapours from unburned fuel is an important driver for contrail ice formation in the low-soot regime at cruise[48]. Furthermore, the effects of engine-to-engine variability (Fig. 4b) and of engine deterioration on particle emissions and contrails need to be investigated. Nevertheless, our results give data-driven information to engine manufacturers for the design of the oil venting system as a lever to reduce total particle emissions.

More experiments are required to investigate contrail formation with sulfur-free fuel to test whether sulfur-free fuels could further reduce contrail ice crystal numbers in the lean-burn mode. The first indications were found for RQL combustors that low-sulfur fuels might reduce

soot particle activation and thereby contrail ice crystal numbers[3,21,32]. Our findings point to a strong effect of low-sulfur fuels in reducing contrail ice crystal numbers in the lean-burn mode. Given that current regulations on Jet-A1 fuel composition with a maximum limit of 3,000 ppmm FSC are more than 30 years old, updates are probably worth considering.

The results of this work need to be implemented in global fleet models. Today, more than 10% of passenger flight distance globally is operated with lean-burning engines, and the trend is increasing[48]. Assumptions of low contrail ice crystal numbers by current lean-burn engines can underestimate their climate impact[7,20,35] and a comprehensive observation-based modelling approach evaluated with data on microphysical and optical contrail properties along lifetime, and atmospheric conditions is required to derive reliable contrail climate forcing predictions. Our results also highlight the importance of reliable assumptions on volatile and non-volatile particle emissions when assessing pathways for climate-neutral aviation, including hydrogen-based technologies[36]. In sum, our findings provide hints for the design of next-generation engines and fuels to progress towards clean and competitive future aviation.

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

# Methods

## Source aircraft and engines

The A321neo Airbus A321neo-251NX (serial no. 7877) was equipped with two CFM LEAP-1A engines. The CFM LEAP-1A35 turbofan engine has a maximum rated thrust of 143.1 kN, a maximum overall pressure ratio of 38.5 and a bypass ratio of 10.5; see Unique Identification Number 01P20CM135 for its emission certification data[29].

The CFM LEAP-1A features a lean-burn combustor or staged combustor with a rich-burn pilot stage and a lean-burn main stage[28]. The lean-burn mode is operated during take-off, climb and cruise phases. Both the central pilot injector and the annular main injector ring inject fuel into the combustion chamber, resulting in wide areas of lean fuel-to-air ratios and a more homogeneous temperature distribution in the combustor. The annular main injector is switched off for descent and taxi phases (rich-burn mode) to avoid combustor instability. Operating conditions typical for cruise were selected for the flight tests, and the T30 temperature at the combustor inlet was fixed for the different measurement points to allow comparability between lean-burn and rich-burn conditions. As the combustor normally operates in lean-burn conditions at cruise, engine steering control adjustments were made by the manufacturer to enable controlled operation in the forced rich-burn mode at cruise.

## Falcon instrumentation

Contrail ice particles, aerosols and trace gases were measured with a set of well-characterized instruments that have been deployed aboard aircraft in previous campaigns[21,27,39,51]. Temperature and other meteorological data were measured with the meteorological measurement system on Falcon[52]. In the following, we describe the instruments and data evaluation used for this study in more detail.

## Contrail ice particle instrument

Contrail ice particles in the size range between 0.6 μm and 50 μm diameter were measured with the Cloud and Aerosol Spectrometer (CAS)[39,53,54], mounted in the inner left underwing pylon of the DLR Falcon. When the Falcon aircraft flies through contrails, ice particles pass through the instrument and scatter light from a laser beam ($\lambda = 658$ nm) in a sample area of $0.22 \pm 0.04$ mm$^2$. By detecting the scattered light intensity, ice particle number concentrations as well as PSDs can be determined using Mie scattering theory and following the calibration method in ref. 55. Ice particle number concentrations are corrected for coincidence effects using an empirically derived correction function described in ref. 39. Shattering effects[56,57] were not observed and, therefore, no correction was performed. The overall ice particle number concentration uncertainty is determined by uncertainties in the use of total air speed for the sample air speed, by the sample area uncertainty and by the concentration-dependent counting uncertainty. This amounts to an overall ice particle number concentration uncertainty of ±20% for the presented measurements.

## Aerosol instruments

Total and non-volatile particle number concentrations were measured using condensation particle counters (CPC) TSI models 3010 and 3768a (TSI), which are modified and optimized for airborne applications. The CPCs show different lower size cut-offs of 5 nm diameter for total particles and 14 nm diameter for non-volatile particles[5]. Aerosol instruments retrieved the sample air through a forward-facing, near-isokinetic inlet. To determine non-volatile particle concentrations, three CPCs were operated behind a heated inlet line of a thermal denuder at 250 °C, removing volatile components. The sample flow could be diluted by about a factor of 30 using an inline dilution system to prevent saturation of the particle counters. CPC data were corrected for reduced detection efficiencies at low pressures and for particle losses in the thermodenuder[5].

Uncertainties in particle number concentration measurements are mainly caused by uncertainties of the low-pressure correction functions, which amount to 7–13% at ambient pressures of 250–350 hPa and may vary slightly between the two CPCs. In the emission index uncertainty analysis, additional contributions arise from aerosol and $CO_2$ background variability, the uncertainty of the $CO_2$ measurement, and of the hydrogen-to-carbon ratio of the fuel. Inlet effects are negligible owing to the small particle sizes (<0.1 μm). Overall, the uncertainty of the particle emission index in the near-field at 300 hPa amounts to about 10%. Further details on particle measurement uncertainties are provided in refs. 5,33.

## Trace gas instruments

Carbon dioxide ($CO_2$) was measured using a high-frequency (10 Hz) non-dispersive infrared gas analyser (LI-7000, LI-COR biosciences) aboard the Falcon. Furthermore, a specifically adapted cavity ring down spectrometer G2401-m (Picarro), well-known for its stable instrument performance, was used to monitor and cross-check $CO_2$ background values[58]. The sample air was passed to both instruments by backwards-facing inlets mounted on the upper part of the fuselage of the Falcon. The LI-7000 was modified in-house for aircraft deployment and makes use of the absorption of infrared radiation by $CO_2$ molecules inside a measurement cell. By comparing the signal with that from a reference cell containing zero air, the absolute absorption and $CO_2$ mixing ratio is derived. An occurring temperature drift of the instrument with time is compensated for by frequent zero measurements every 30 min during the flight. In the post-processing, the $CO_2$ mixing ratio is corrected for water vapour dilution to report dry-air mole fractions. The accuracy of the LI-7000 $CO_2$ measurement is 0.2 ppm, thereby taking account of the reproducibility of the calibration standards (0.18 ppm; using NOAA standards traceable to the WMO $CO_2$ calibration scale), the precision (0.08 ppm) and the uncertainty of water vapour dilution correction (0.1 ppm) (for more details, see ref. 58).

Reactive nitrogen ($NO_y$) was measured using a chemiluminescence detector (CLD TR 780, ECO PHYSICS)[58]. Using a heated gold converter ($T = 290$ °C) with hydrogen ($H_2$) as a reducing agent, reactive nitrogen species $NO_y$ ($NO + NO_2 + HNO_3 + PAN$ and others) are converted to NO molecules, which subsequently go through the chemiluminescence reaction with $O_3$. Owing to the measurement range of the instrument of up to 1,000 ppbv, a dilution system was integrated to measure higher concentrations in the exhaust plumes. Thereby, zero air was added to the sample air at a ratio of 1:4 before the measurement. The detection limit of the CLD TR 780 is 0.55 ppbv at a time resolution of 1 Hz (ref. 58).

Water vapour ($H_2O$) mixing ratios were measured with the water vapour mass spectrometer AIMS-$H_2O$ (refs. 59,60) with a frequency of 2.5 Hz by sampling air through a backwards-facing inlet at the upper fuselage of the Falcon. The relative humidity over ice ($RH_i$) as a relevant parameter for contrail persistence has been calculated from measurements of the water vapour mixing ratio, static air temperature and static pressure. For the calculation in this study, the ice saturation pressure formulation from equation 7 in ref. 61 was used. The uncertainty in $RH_i$ is estimated to be around 15% by error propagation of the uncertainty in water vapour mixing ratios (8–12%) and temperature (0.5 K). Uncertainty in static pressure has only a minor contribution to the $RH_i$ uncertainty.

## Meteorological dataset

The aircraft is equipped with basic instrumentation that measures pressure, temperature, air flow, wind speed and humidity at data rates of up to 100 Hz. The quality of the measurement depends not only on a proper lab calibration of the sensors but also on an accurate parameterization of the aerodynamic effects in the vicinity of the aircraft fuselage. These effects were determined by in-flight calibration methods, including the trailing cone method and manoeuvres[62].

## The NEOFUELS/VOLCAN campaign

Fifteen flights were performed with the Falcon behind the A321neo over the Atlantic and the Mediterranean in March 2023. For the six emission flights, the A321neo and the DLR Falcon entered a two-aircraft formation with a constant air speed of 0.59 Mach at an altitude of 9–10 km. Starting at a close distance of about 40 m to the right-hand engine of the leading aircraft, the Falcon repeatedly entered the emission plume of the A321neo from below, acquiring plume emission data for 45 s, followed by a 30-s sequence of background data, with five repetitions at the same engine setting to obtain good data statistics. As the Falcon was pushed back by the exhaust plume during the sampling sequences, the distance of the Falcon from the engine exit plane increased. To avoid entrainment of the wing-tip vortices, the Falcon descended below the plume at about 250 m distance and increased speed to catch up with the Airbus. This measurement sequence was repeated several times under different engine conditions or for different fuels in most sampling sequences, except one probing the right engine. Contrails are not yet fully developed at these close distances, and therefore contrails were probed in the far field at distances of 6–29 km behind the A321neo aircraft flying at a mean typical cruise speed of 0.78 Mach with both engines operating at the same combustor inlet temperature T30, combustor settings and powered by the same fuel.

## Engine emission measurements and evaluation

A time series of data collected by the DLR Falcon during a near-field emission measurements flight on 7 March 2023 is shown in Extended Data Fig. 1. Encounters of 0.2–1.4-s-old A321neo LEAP-1A exhaust plumes are evident by repeated sequences with large enhancements in total particles, $CO_2$, $H_2O$ and $NO_y$ concentrations, and temperatures above background levels. Smaller fluctuations in the flight altitude indicate direct pilot manoeuvres in the exhaust plume with the auto-pilot switched off. Here, the pilots split aircraft control, and one pilot steered and kept the Falcon in the exhaust, whereas the other operated the thrust. A sequence of five plume encounters at the same engine power and combustor settings is followed by a break of few minutes to measure ambient conditions and to allow the change of engine power settings with respect to T30 combustor inlet temperature, combustion mode or fuel. Several orders of magnitude enhancements in total particle concentrations ($d > 5$ nm) are measured in all plume sequences, whereas distinct different features are observed in the non-volatile particles. Similar to total particles, large peaks are observed in non-volatile particle concentrations in the exhaust in the forced rich-burn mode, whereas non-volatile particle concentrations are close to background levels in the lean-burn combustion mode at similar T30 combustor inlet temperature settings.

## Calculation of emissions indices

To conduct valid comparisons of particle number concentrations independent of dilution level, particle number concentration enhancements $\Delta X$ need to be compared with the mixing ratio enhancement of a tracer such as $CO_2$ ($\Delta CO_2$). By assuming homogeneous mixing of particles and trace gases, the resulting ratio $\Delta X/\Delta CO_2$ serves to gauge the level of plume/contrail dilution[4]. The amount of emitted $CO_2$ per mass of burned fuel is a fuel property, depending on the ratio of hydrogen to carbon atoms in the fuel, and is described by the $CO_2$ emission index $EI_{CO_2}$. With the ratio of molar mass of air ($M_{air}$) to molar mass of $CO_2$ ($M_{CO_2}$) and density of air ($\rho_{air}$), an emission index for species $X$ can be calculated as[4]

$$EI_X = \left(\frac{\Delta X}{\Delta CO_2}\right) \cdot \left(\frac{M_{air}}{M_{CO_2} \cdot \rho_{air}}\right) \cdot EI_{CO_2}$$

Here, density of air and particle concentration enhancement $\Delta X$ are given at standard temperature and pressure for the aerosol measurements and at ambient conditions for ice particle measurements. For non-volatile and total particles, $EI_X$ describes the number of emitted particles per mass of fuel burned, $EI_{nv}$ that of non-volatile particles and $EI_t$ that of total particles, including volatiles and non-volatiles. Ice particles are not directly emitted by the engine and form on emitted or freshly nucleated aerosols in the young plume; they are labelled $EI_{ice}$ for consistency with $EI_{nv}$ and $EI_t$.

Contrail encounters are evaluated only when ice particle and trace gas measurements are conducted approximately simultaneously. Therefore, correlations of ice particle concentration and $CO_2$ mixing ratio time series are calculated, and contrail encounters are rejected if the resulting correlation is lower than 0.6, similar to the method in refs. 27,39. Uncertainties in aerosol/ice particle measurement, $CO_2$ and aerosol particle background determination, $CO_2$ measurement and ambient condition measurements are propagated to determine an EI uncertainty for every plume–contrail encounter. This results in average $EI_{nv}$ and $EI_t$ uncertainties of $10 \pm 5\%$ and an average $EI_{ice}$ uncertainty of $38 \pm 16\%$.

In contrast to emission measurements performed in ice-free conditions at close distance, contrails were probed 6–29 km further downstream. At those distances, the contrail is in a relatively stable state if ambient conditions are supersaturated with respect to ice, so that valid comparisons of combustion modes can be performed. Contrail ages are determined from the GPS positions of the two aircraft and using wind field measurements onboard the preceding aircraft to determine contrail drift[39]. Ambient conditions, as well as fuel properties, and an assumed overall propulsion efficiency of 0.36 (ref. 63) enable calculation of the Schmidt–Appleman contrail formation threshold $T_{SA}$ (ref. 50). The difference in the ambient temperature to the formation threshold $\Delta T_{SA}$ ranged between $-4.5$ K and $-19.0$ K for the shown contrail encounters. Extended Data Tables 2–4 show the engine and ambient conditions for contrail data shown in Figs. 2–4. As volatile and non-volatile particle emission data during the contrail sampling events could have been spoiled by the presence of contrail ice crystals, the engine particle emission data for Figs. 2, 3 and 4 were taken from near-field emission measurements at the same engine conditions in terms of engine inlet temperature T30, and the same fuels. Slightly different engine conditions or fuel contaminations also explain the lower $EI_{nv}$ for Jet A-1 compared with engine emission data shown in Fig. 1 and Extended Data Table 1.

## Contrail particle size distributions

Apart from the changes dependent on fuel and engine mode in contrail ice particle numbers, we investigate variations in contrail PSDs. To this end, PSDs of single contrail encounters formed on emissions from Jet A-1 and the HEFA-blend are shown in lean-burn and rich-burn combustion modes in Extended Data Fig. 2 for ambient measurement conditions given in Extended Data Table 2.

The contrail PSDs for each fuel type and engine mode may exhibit a slight dependence on initial ice crystal numbers because of changes in fuel type and combustion mode. Lower initial ice crystal numbers, for example, for the HEFA-blend compared with Jet A-1 in the forced rich-burn mode, can lead to slightly larger ice crystal diameters, as the water vapour emitted by the engines is distributed onto fewer ice crystals for the HEFA-blend. A similar trend has been observed in ref. 21 for a semi-synthetic jet fuel compared with Jet A-1 in a contrail analysis for similar ambient conditions, contrail age and at a similar difference to the emission altitude. Changes in water vapour and $RH_i$ during ice crystal growth may also affect microphysical and optical contrail properties, which may have an effect on the radiative forcing of the contrail.

## Aerosol and contrail microphysics model

The ACM model in ref. 3 is further developed and improved for this study. The ACM model is a parcel model of jet plume aerosol and ice microphysics developed in the late 1990s (refs. 43,64,65), with the

volatile particle formation module improved with algorithms and thermodynamic data developed in the past two decades[40,42], and the contrail microphysics module improved with a new soot activation scheme[3,31]. Thereby, the fraction of fuel sulfur oxidized to sulfuric acid during combustion assumed by ACM is 3% (ref. 3). The ACM model captures the dependence of contrail ice particles formed on emitted non-volatile soot particles and can explain less-than-unity fractions of soot particles forming contrail ice particles[3] as observed for low-sulfur fuels during ECLIF campaigns[21,27]. A recent study can also explain observed higher contrail ice crystal numbers than soot particle emissions, through additional activation of volatile sulfate aerosol in the soot-rich mode[33]. More importantly, previous ACM simulations have predicted that the number of contrail ice particles formed when soot emission is very low (that is, in the soot-poor regime) can be comparable to those of the soot-rich regime because of the activation of volatile particles[2,3,31]. More details of the ACM model can be found in ref. 3.

For the present study, the model has been improved to simulate the competition between nucleation and condensation of sulfuric acid, as well as organic and lubrication oil vapours, with individual nucleation and condensation rates assigned to each species. The ACM explicitly calculates the nucleation and condensation of lubrication oil vapours with an emission index ($EI_{oil}$) of 120 mg kg$^{-1}$ fuel. $EI_{oil}$ is informed by oil consumption data released by the engine manufacturer of 110–140 mg kg$^{-1}$ fuel for the tested LEAP-1A engine, comparable to recent ground measurements[41,47] for different engines of 110 mg kg$^{-1}$. $EI_{oil}$ is also adjusted to our near-field particle emission measurements behind the left and the right engine[39] (Fig. 4). About 6–7% of $EI_{oil}$ may lead to new particle formation, corresponding to the ULVOC fraction of lubrication oil reported in ref. 40, and the rest is assumed to condense on pre-existing aerosol.

Also, the nucleation of low-volatile organic hydrocarbons from incomplete fuel combustion is considered, with an emission index $EI_{org}$ of 50–66 mg kg$^{-1}$ fuel estimated from ground to cruise projections. However, only a fraction of those organic vapours emitted by the engines—containing smaller organic molecules than lubrication oil vapours[41,47]—may lead to new particle formation. Therefore, the assumed $EI_{org}$ in the ACM simulation is lower, around 5 mg kg$^{-1}$ fuel. In the model, $EI_{org}$ may add to new particle formation with nucleation rates in the range of sulfuric acid nucleation, and different from the nucleation rates of the very low-volatile lubrication oils, or may condense on existing aerosol.

Owing to an overnight leakage between the two tank systems, the original 100% HEFA-SPK was contaminated and blended with Jet A-1 for the data shown in Fig. 2, which may have changed the FSC of the Jet A-1 to 192 ppmm and led to the HEFA-SPK blend with 75 ppmm sulfur. In the rich-burn combustion mode, the model simulates a decrease in soot and contrail ice particles in agreement with emission and contrail observations for low-aromatic fuels and RQL engines[21,27,39]. Slightly lower $EI_{nv}$ than $EI_{ice}$ for the forced rich-burn mode in our study might point to an additional activation of smaller soot particles, below the detection limit of the particle counter, or of volatile aerosols contributing to ice nucleation, which are not yet completely covered by the model. In the lean-burn combustion mode, the model simulates a decrease in contrail ice crystal numbers for fuels with a lower sulfur content and for increasing ambient temperatures, generally consistent with the observations within the experimental and model uncertainties. Existing deviations from the observations might point to additional factors or processes not yet covered by the model.

### Sensitivity study for organic vapours

The Modèle Microphysique pour Effluents (MoMiE) is a microphysical model first developed at ONERA by Sorokin et al.[66] and Vancassel et al.[67] for typical kerosene fuels, and then adapted to simulate the combustion of SAF[34]. The model accounts for heterogeneous nucleation on soot particles and homogeneous nucleation of volatile particles composed of sulfur and organic species[68]. The model further differentiates between two types of organic species: water-soluble organic compounds that can nucleate to form a new volatile aerosol distribution and water-insoluble organic compounds that can condense on soot and volatile particles[69]. The processes of coagulation, condensation and freezing[70], as well as the effects of ion recombination by considering positive organic clusters and negative sulfates[71,72], are included. Plume dilution is calculated using the analytic formula in ref. 50.

The MoMiE simulations span a range of initial soot particle number emission indices and consider three fuel types: Jet A-1, HEFA-blend and 100% HEFA-SPK (Table 1, Extended Data Table 2 and Extended Data Fig. 3). In all simulations, soot particles are represented by a lognormal size distribution with 35 nm median diameter and a standard deviation of 1.6 (ref. 3). The emission index of the insoluble organic compounds from condensing oil vapours, $EI_{oil}$ is fixed to 120 mg kg$^{-1}$ fuel, in line with data from the engine manufacturer for this engine model. $EI_{oil}$ does not produce new particles in this model, but can condense on existing soot and volatile particles. Ground to cruise extrapolations suggest $EI_{org}$ in the range of 50–66 mg kg$^{-1}$ fuel, and the model assumes about 10% to be converted into volatile aerosol, which is activated to ice. Extended Data Fig. 3 provides simulation results for Jet A-1, HEFA-blend and HEFA-SPK scenarios with different fuel sulfur contents (192 ppm, 75 ppm and 3 ppm, respectively, by mass of fuel sulfur; Table 1) simulated for average ambient conditions of 218 K and 110% relative humidity with respect to ice, compared with the observations of Jet A-1 and HEFA-blend (Extended Data Table 2). Extended Data Fig. 3 shows the influence of initially soluble organic compounds on new particle formation, complementing sulfuric acid-driven particle nucleation. The simulated ice particle numbers are sensitive to low-volatile organic vapours in the low-soot regime. Increasing water-soluble organic vapour concentrations enhances contrail ice crystal numbers in the low-soot regime for both measurement cases and yields a better agreement with the observations for the Jet A-1 case. In the rich-burn combustion mode, the presence of organics has a smaller influence on ice particles, and soot particles are the main contrail ice nuclei.

### Data availability

The datasets for this study are available in the data repository at https://halo-db.pa.op.dlr.de/ with the following https://doi.org/10.17616/R39Q0T. Data are indexed as follows: #10498: Extended Data Fig. 1; #10580: Fig. 1 and Extended Data Table 1; #10580 + #10581 + #10582: Figs. 2–4 and Extended Data Tables 2–4; #10584–10592: timeseries of measurement data; #10593–10601: plume encounter definitions; and #10937–10945: contrail ice PSDs for Extended Data Fig. 2. See also https://doi.org/10.25358/openscience-11418, for details on data evaluation.

### Code availability

The aerosol and contrail microphysics (ACM) model is the proprietary intellectual property of the University at Albany, State University of New York (SUNY-Albany) and is not publicly available. As such, there is no non-proprietary code. For those wishing to use the proprietary code, a software licensing agreement with specifics depending on the nature of usage needs to be signed to use the model. Parties interested in using the model should contact the Office of Technology Transfer of SUNY-Albany at https://www.albany.edu/research-economic-development/contact#technology-transfer or UAOIDC@albany.edu or pgonczlik@albany.edu. The values of model curves in Figs. 2 and 3 are available in the data repository at *Zenodo* (https://zenodo.org/records/18830350). The Modèle Microphysique pour Effluents (MoMiE) model has been developed by and is the proprietary intellectual property of ONERA, France, and is not publicly available. The use of the MoMiE code requires research collaboration

within the framework of a contractual partnership (contact margaux.vals@onera.fr or nicolas.bonne@onera.fr). The values of model curves in Extended Data Fig. 3 are available in the data repository at Recherche Data Gouv V (https://doi.org/10.57745/WAQGKH).

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

**Acknowledgements** We thank the flight crews of the DLR Falcon and the Airbus A321neo for excellent flight operations. This work was supported by the DLR Aeronautics Research Programme within the Project NEOFUELS, and by the French VOLCAN programme funded by the Direction Générale de l'Aviation Civile (DGAC). C.V., G.N. and T.B. acknowledge funding by DFG by projects nos. 510826369 (ECOCON), 316646266 (HALO), 442649092 (MICRO-ICE), and 428312742 (CRC 301), and by the Horizon Europe program of the European Union under grant no. 101192301 (A4CLIMATE). F.Y. acknowledges funding support from GE Aerospace, Simons Foundation International (SFI) (SFI-MPS-SRM-00012042), and the US National Science Foundation (NSF) (AGS-2325458).

**Author contributions** C.V., C.R., S.K. and K.S. planned and coordinated the flight experiment. R.M., D.S., R.D., S.K., T.B., T.J.-W., L.E., C.H., E.H., M.L., A.M., G.N., A.Roi., M.S., P.S. and A.G. performed the in-flight measurements and analysed the data. G.L.C., J.M. and E.R.-E. operated the A321neo and the instruments. J.Z. and A.Roc. assisted with engine data interpretation. K.S., G.E. and P.L.C. analysed the fuel compositions. R.M. performed the contrail data evaluation, R.D. the aerosol evaluation and T.B. the trace gas analysis. A.D. performed the contrail forecast. F.Y. further developed the ACM model and performed model runs. N.B. and M.V. applied the MoMiE model and performed model runs. C.V. wrote the paper. All authors contributed to the paper.

**Funding** Open access funding provided by Deutsches Zentrum für Luft- und Raumfahrt e.v. (DLR).

**Competing interests** The authors declare no competing interests.

**Additional information**
**Correspondence and requests for materials** should be addressed to Christiane Voigt.

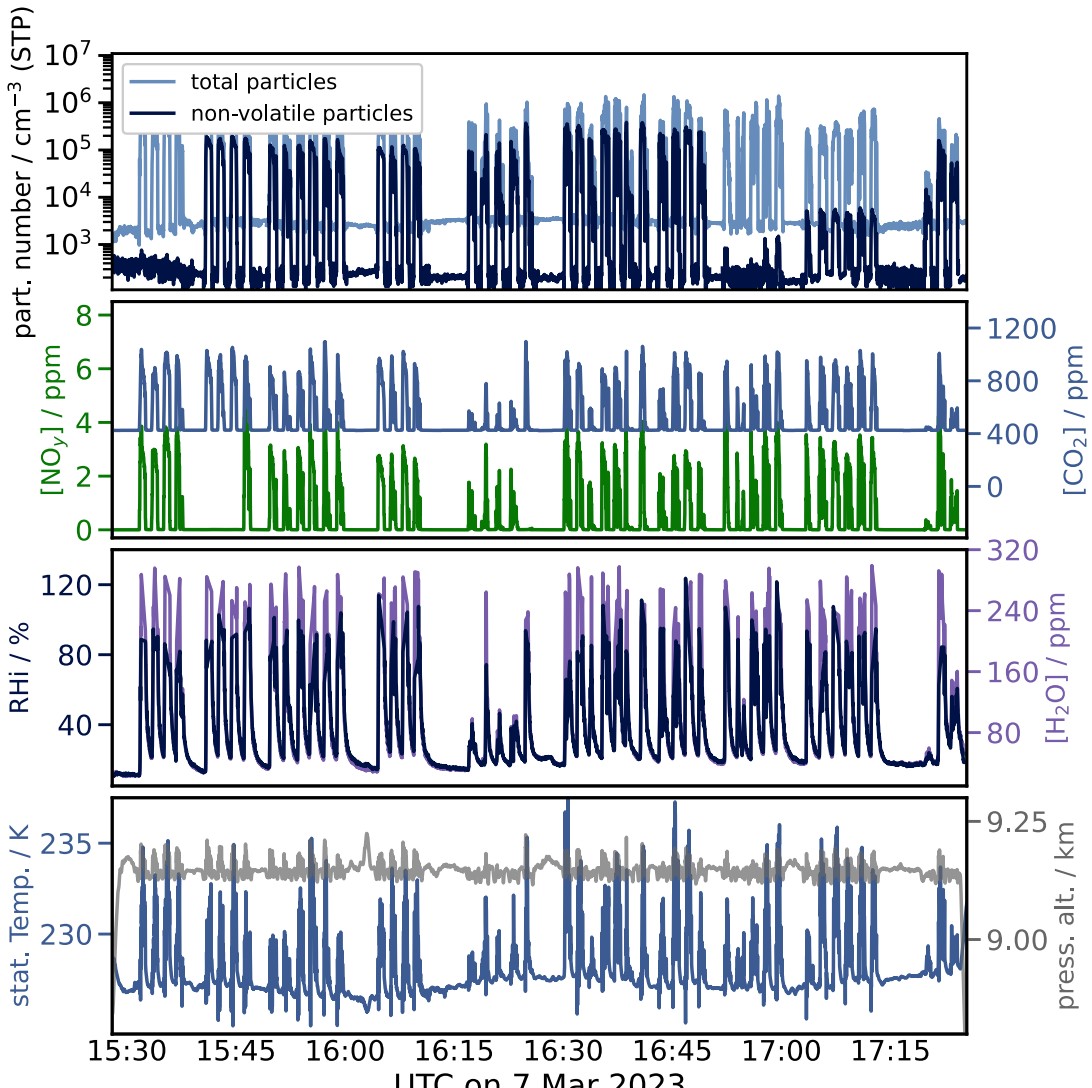

**Extended Data Fig. 1 | Time series of trace gas and particle data measured on 7 March 2023 in the A321neo exhaust plume during forced rich-burn, split point and lean-burn measurement sequences.** Number concentration of non-volatile particles (i.e. soot, d > 14 nm, dark blue line) and of total (d > 5 nm, light blue line) particles, i.e., volatile and non-volatile particles in cm⁻³ at standard pressure, $CO_2$ (blue), $NO_y$ (green), and $H_2O$ (purple) mixing ratio in parts per million by volume, relative humidity with respect to ice (RHi, blue) in percent, static temperature (blue) in Kelvin and pressure altitude (light grey) in kilometres. Plume measurements were taken in the 0.2 to 1.4 sec-old plume at distances between 40 and 250 m. Measurements in the lean-burn mode are visible by low soot particle concentrations, e.g. at 15:30 to 15:40 UTC or 16:50 to 17:00 UTC (see Fig. 1). The intermediate fuel split point with slightly enhanced non-volatile particles was measured at 17:03 to 17:13 UTC.

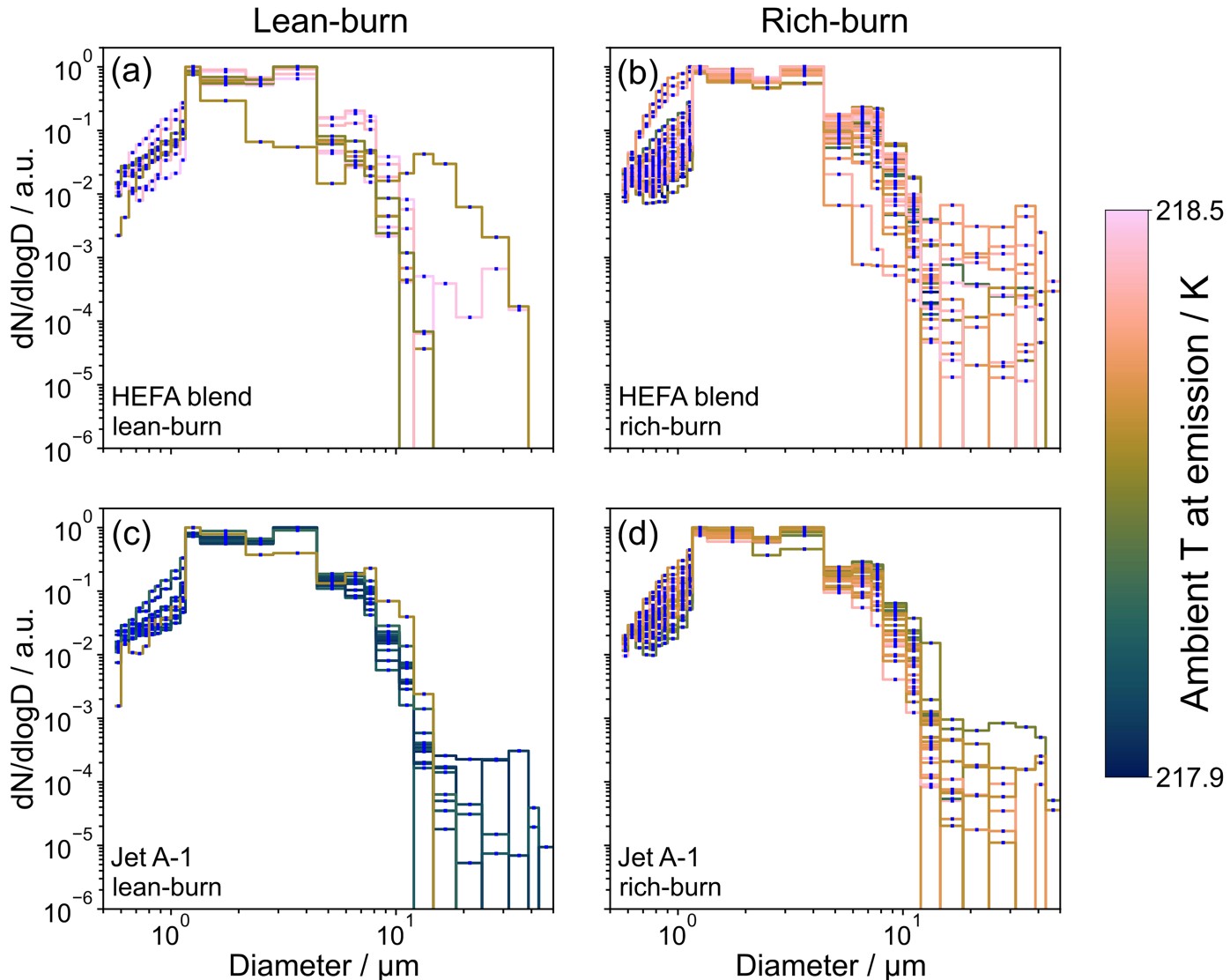

**Extended Data Fig. 2 | Contrail ice particle size distributions of contrail encounters in lean-burn and rich-burn engine configurations for Jet A-1 and HEFA-blend.** The colour bar shows the ambient temperatures at contrail formation. PSDs of ice number concentrations per logarithmic bin width were normalized to the bin with the highest concentration to illustrate trends in ice crystal sizes - and do not reflect absolute number concentrations. Ambient conditions are given in Extended Data Tables 2 and 3. Mean effective diameters for HEFA-blend are (**a**) 4.5 μm in lean-burn mode and (**b**) 5.1 μm in rich-burn mode, and for Jet A-1 (**c**) 4.3 μm in lean-burn mode and (**d**) 4.6 μm in rich-burn mode. Contrail ice PSD data were not filtered for correlation and uncertainty as mentioned in Extended Data Table 3, and mean Jet A-1 lean-burn contrail RHi is 111%. All other ambient conditions in Extended Data Table 2 apply.

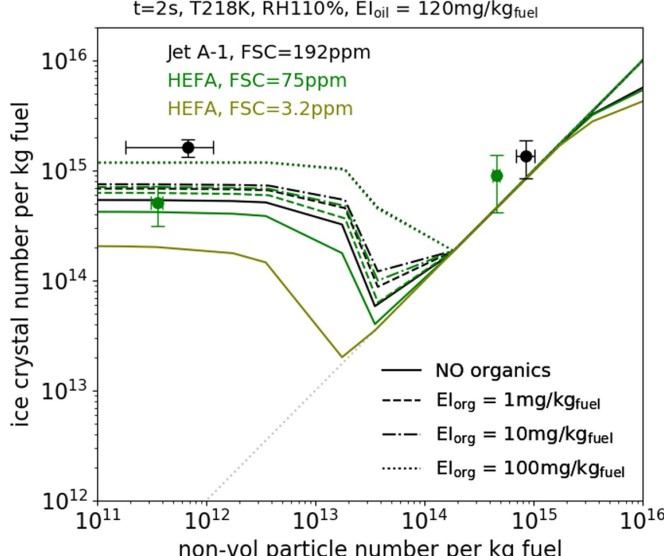

**Extended Data Fig. 3 | Correlation between non-volatile (i.e. soot) particle number emission index and the ice emission index, and model sensitivity to the presence of organic vapours.** Symbols show observations (Extended Data Table 2) at similar atmospheric conditions in terms of temperature and RHi as black (Jet A-1) and green (HEFA-blend) circles. Lines show results from the updated contrail microphysical model[34] simulated for 218 K for Jet A-1 (solid black line, 192 ppmm sulfur) and two HEFA-SPKs (solid green and olive lines) with different fuel sulfur contents (75 and 3 ppmm sulfur). Results of modelled emission indices for no organic vapours (solid lines) and of emission indices for water-soluble organics $EI_{org}$ of 1, 10 and 100 mg kg$^{-1}$ fuel are shown as dashed, dash-dotted and dotted lines, respectively. For Jet A-1, the model is in better agreement with the observations when $EI_{org}$ and $EI_{oil}$ are considered.

**Extended Data Table 1 | Particle emission indices for Jet A-1 fuel for different combustion modes and ambient parameters**

| Combustion mode | | Lean-burn | Forced rich-burn |
|---|---|---|---|
| **Plume age (s)** | | 2-3 | 2-3 |
| **Sampling time (s)** | | 215 | 850 |
| **Altitude source aircraft (m)** | | 9140 | 9140 |
| **Speed source aircraft (Mach)** | | 0.59 | 0.59 |
| **Ambient RHi (%)** | | 68-90 | 40-86 |
| **Ambient T (K)** | | 229-231 | 228-232 |
| **Delta T_SA (K)** | | 0.5-1 | 0.5-2 |
| **$EI_{nv}$ (kg$^{-1}$ fuel)** | **Median** | $1.0 \times 10^{12}$ | $1.0 \times 10^{15}$ |
| | **Range** | $(0.5\text{-}1.9) \times 10^{12}$ | $(0.8\text{-}1.1) \times 10^{15}$ |
| **$EI_t$ (kg$^{-1}$ fuel)** | **Median** | $2.1 \times 10^{15}$ | $2.7 \times 10^{15}$ |
| | **Range** | $(2.0\text{-}3.3) \times 10^{15}$ | $(2.1\text{-}4.0) \times 10^{15}$ |

Emission data (median and range), related aircraft data and ambient conditions for the flight on 7 March 2023 shown in Fig. 1.

**Extended Data Table 2 | Non-volatile and ice particle emission indices for fuels and combustion modes measured at similar ambient conditions**

| Fuel type / combustion mode | mean RHi [%] | mean $T_{amb}$ [K] | $p_{amb}$ [hPa] | FSC [ppmm] | $EI_{H2O}$ [kg kg$^{-1}$ fuel] | $EI_{nv}$ [kg$^{-1}$ fuel] | $\Delta EI_{nv}$ [kg$^{-1}$ fuel] | $EI_{ice}$ [kg$^{-1}$ fuel] | $\Delta EI_{ice}$ [kg$^{-1}$ fuel] |
|---|---|---|---|---|---|---|---|---|---|
| **Jet A-1 / lean-burn** | 112 | 218 | 300 | 192 | 1.26 | $6.71\times10^{11}$ | $4.89\times10^{11}$ | $1.63\times10^{15}$ | $3.06\times10^{14}$ |
| **Jet A-1 / rich-burn** | 112 | 218 | 300 | 192 | 1.26 | $8.52\times10^{14}$ | $1.68\times10^{14}$ | $1.36\times10^{15}$ | $5.06\times10^{14}$ |
| **HEFA-blend/ Lean-burn** | 105 | 218 | 300 | 75 | 1.33 | $3.56\times10^{11}$ | $4.31\times10^{10}$ | $5.14\times10^{14}$ | $2.03\times10^{14}$ |
| **HEFA-blend/ rich-burn** | 111 | 218 | 300 | 75 | 1.33 | $4.57\times10^{14}$ | $3.15\times10^{13}$ | $9.06\times10^{14}$ | $4.86\times10^{14}$ |

Ambient conditions, fuel sulfur content, water vapour ($EI_{H2O}$), mean non-volatile ($EI_{nv}$) and mean ice particle emission indices ($EI_{ice}$) and their standard deviation for data from Jet A-1 and HEFA-blend fuels in forced rich-burn and lean-burn combustion modes shown in Fig. 2 and Extended Data Figs. 2 and 3. Contrail data were measured in selected domains with similar atmospheric conditions (relative humidity with respect to ice, ambient temperature, and pressure)[39]. As emission data may have been perturbed during contrail encounters, $EI_{nv}$ were taken from near-field emission measurements at similar engine conditions.

**Extended Data Table 3 | Ice particle emission indices and ambient conditions for contrail data**

| Combustion mode | Lean-burn | Split | Forced rich-burn |
|---|---|---|---|
| **Contrail age (s)** | 29-117 | 36-65 | 24-122 |
| **Sampling time (s)** | 1882 | 604 | 3673 |
| **Ambient RHi (%)** | 100-144 | 100-123 | 100-143 |
| **Ambient T (K)** | 211-223 | 214-218 | 210-223 |
| **Delta $T_{SA}$ (K)** | -4.5 to -13.9 | -7.9 to -12.5 | -5.5 to -14.9 |
| **Altitude source aircraft (m)** | 9050-11488 | 9646-10608 | 9047-11490 |
| **Speed source aircraft (Mach)** | 0.752-0.794 | 0.745-0.793 | 0.641-0.798 |
| **$EI_{ice}$ range (kg$^{-1}$ fuel)** | $(0.06\text{-}2.4) \times 10^{15}$ | $(0.2\text{-}1.8) \times 10^{15}$ | $(0.05\text{-}3.3) \times 10^{15}$ |

Aircraft and ambient conditions and range of contrail $EI_{ice}$ shown in Figs. 3 and 4, filtered for 100% RHi, >60% correlation of ice particle concentration and tracer, <100% uncertainty, and stable engine conditions (see also ref. 39).

**Extended Data Table 4 | Ambient conditions and range of particle number emission indices for the same engine conditions as contrail data**

| Combustion mode | | Lean-burn | Split | Forced rich-burn |
|---|---|---|---|---|
| Plume age (s) | | 0.2-1.4 | 1 | 0.2-1.4 |
| Sampling time (s) | | 1828 | 343 | 1811 |
| Ambient RHi (%) | | 40-80 | 60-80 | 40-80 |
| Ambient T (K) | | 228-235 | 228-231 | 228-235 |
| Delta $T_{SA}$ (K) | | 0.5 - 5 | 0.5 | 0.5 - 5 |
| Altitude of source aircraft (m) | | 8830-9140 | 9140 | 8830-9140 |
| Speed of source aircraft (Mach) | | 0.59 | 0.59 | 0.59 |
| $EI_{nv}$ (kg$^{-1}$ fuel) | Median | $4.6 \times 10^{11}$ | $1.6 \times 10^{13}$ | $8.5 \times 10^{14}$ |
| | Range | $(1.1-6.3) \times 10^{11}$ | | $(3.8-9.5) \times 10^{14}$ |
| $EI_{t}$ (kg$^{-1}$ fuel) | Median | $1.3 \times 10^{15}$ | $1.1 \times 10^{15}$ | $1.5 \times 10^{15}$ |
| | Range | $(0.4-3.1) \times 10^{15}$ | | $(0.8-2.5) \times 10^{15}$ |

As volatile and non-volatile particle emission data during the contrail sampling events could have been spoiled by the presence of contrail ice crystals, the particle emission data for Figs. 2 to 4 were taken from near-field emission measurements at the same engine settings in terms of engine inlet temperature T30; see Methods.