## [Peer Review File · Nature]

Substantial aircraft contrail formation at low soot emission levels

Corresponding Author: Professor Christiane Voigt

Co-reviewers 3 and 4 were unable to review the first revision. We therefore added reviewer 5 and asked them to assess the authors' replies and revisions to the now-missing co-reviewers.

Version 0:

Reviewer comments:

Referee #1

(Remarks to the Author)

General Comments:

This is an important, ground-breaking study that shows strong evidence of contrail formation on volatile particulate matter (vPM). This is an exciting and significant finding, since this demonstrates that, even for aviation engines that emit essentially no soot/non-volatile PM (nvPM), contrails can still form. This has not been shown before and provides important new insight into aviation contrails, which are one of the largest climate impacts from aviation. This should be of great interest to communities that are keeping track of advances in the understanding of anthropogenic climate change and of the contribution of aviation to climate change.

The paper is generally well-written and explains the vPM-contrail connections well. However, attempts to describe the possible effects of fuel composition and ambient conditions on the contrail properties are lacking. If there is data supporting their claims in these areas, that has not been included in the figures or in other ways. So these claims, while interesting and potentially insightful, are not demonstrated in the data presented in this paper. Thus, the claims in the abstract "Our results demonstrate the impact of lean-burn engine configurations, fuel composition, and ambient conditions on contrail ice crystal numbers." should only read "Our results demonstrate the impact of lean-burn engine configurations (and resulting changes in nvPM emissions levels) on contrail ice crystal numbers." based on what is presented. More detailed discussion of why follows.

Figure 2 and correlations between model results and data

Figure 2 plots measured ice particle data versus nvPM number data in several ways. Figure 2a plots those data in comparison to model plots for simulations for different fuel compositions. Yet, the measurement data is not labelled as to what fuel was used for each point. Only the engine configuration is shown by symbol color. While knowing the engine configuration (lean/split/rich) is worthwhile, all points in each vertical grouping is the same engine configuration, so using symbol color is not needed. Each vertical grouping could be labelled by a text box or similar and the data symbols could be instead colored by the fuel type to see if there is a correlation between the data and the model results. As presented, the data could be correlated with fuel type. Or it could be random or it could be anti-correlated. Presumably the authors know what fuel was used for each point, but that is not evident to the reader. So, the plot does not show any correlation between data and model according to fuel type.

Similar arguments apply to Figures 2b and 2c as well. The measurement data is only colored by lean/split/rich and there is no way to determine if the model's variation with organic matter (OM) contribution (2b) or temperature (2c) correlates with the measured data or not. Again, the data could be correlated, random, or anti-correlated based on what is presented in the

Figures. It seems possible that perhaps the wrong versions of the figures were included with the paper, since the authors make the statement “In the lean-burn mode, a strong dependence of EI_ice on ambient temperature is observed and modelled.” Yet that is not shown in the Figure 2c as presented. If the authors have other versions of these figures that actually demonstrate these correlations, they need to present them to make these claims.

I am astounded that, with the long list of coauthors on this paper, the above was not caught in review prior to submission. The statements related to fuel composition (including OM) and temperature are just not supported by what is included in the figures.

But, beyond that, all of the positive discussion of comparisons between model and data ignore some obvious short-comings that are not discussed at all. For the “forced rich-burn” cases, the model results all show a very tight clustering at that range of nvPM ($\sim 10^{15}$). Yet the measured data has as wide a vertical spread for this case as for the other two (lean, split). To say that there is any correlation between models and data is very tenuous. Again, perhaps if the measured data were colored by fuel properties (2a, 2b) or temperature (2c), there might be some correlation for the lean and split cases, but basically there is a wide scatter (\sim one order of magnitude) in the vertical direction for all clusters of points.

The most significant shift in the range and centroid of the data clusters is between the two lean-burn clusters (at $\sim 10^{11}$ and $\sim 10^{12}$) where all the models are totally horizontal with no variation. What is the difference between these two lean-burn clusters (different fuels?)? That seems like the most significant variation amongst the data and it not captured at all by the models. Yet this is not even mentioned in the paper. There is also a smaller shift in range and centroid for the two rich-burn cases, but that might be consistent with the slope of the well-known relation between EI_ice and EI_nvPM in the soot-rich region. But those two rich-burn cases are so close in nvPM that there is probably little new insight from this.

Basically, all these comparisons with the two models’ results add nothing to the paper as presented. The data seems to not depend on whether nvPM is present or not, which is a main conclusion of the paper. That could be shown in a single plot of the data like in Figure 2, but with no model curves included. (i.e., use any of 2a,2b,2c with the model curves removed.) Any conclusions about fuel composition and temperature dependences are not supported by Figure 2 plots. And even if the authors later label the measured data to show some correlation for the lean-burn and/or split cases, they will need to explain why the models disagree so strongly with the data for the forced rich-burn cases, where the model results have no spread and the data has the same spread as for the other cases.

Additionally, and importantly, the model used in Figure 2a (ACM) seems to provide very different results than those for the model used in Figure 2b (MoMie). While MoMie uses OM values of 5 and 50 (as well as 0 and 500), ACM uses a LOM of 25 which might be expected to lie between 5 and 50. MoMie would suggest that FSC of 3 and 195 would lie essentially on top of each other, while ACM shows them a factor of 30 different. There is no discussion of this huge discrepancy between the models. And all of these model calculations are a distraction from the main point that contrails can form on vPM.

A separate paper on model comparisons to this data might make more sense.

Given that:

- 1) the models disagree with each other
- 2) neither model captures the spread behavior for the forced-rich case
- 3) the data are not plotted to show trends with fuel type/organic level/temperature, the data cannot be compared with model results

It seems that the model results should be removed from the paper unless these issues are addressed in the paper.

Low Fuel Sulfur fuel analysis and inclusion of these fuels in plots

The discussion and use of fuel sulfur content (FSC) from Table 1 is confusing. First, while often FSC is reported in %mass, as done here, another common reporting unit is ppm (parts per million by mass) rather than %mass. And when very low FSCs are being explored, ppm is much easier to read rather than dealing with so many preceding zeros. But, more importantly, the table shows entries for 0.5 and 1.9 ppm, while the models and plots only seem to consider the cases 3.2 ppm FSC and higher. Were these very low cases (0.5, 1.9) included in the plots or not? If the low cases are in the plots, why did the models only use 3.2 ppm as the lowest FSC case? And what method was used to measure FSC and what uncertainty was ascribed to these values? One standard FSC method is ASTM D5453, which cannot measure below 1 ppm (LOD 1 ppm, with LOQ ~ 3 ppm[?]). So how was 0.5 ppm FSC measured and what is the uncertainty? (Actually, what is the uncertainty for all low FSC data?) And which data in the plots correspond to these very low FSC values or why was that data not included?

Detailed comments:

Abstract page 1 line 27-28: As stated above, the phrases “fuel composition, and ambient condition” need to be removed unless and until the correlations described in discussion of Figure 2 can be supported by data which might be included in the paper in the future.

Page 7 lines 4-5. While technically not incorrect, the language “fuel sulfur content with EI ranging from 195 (below world average) to 3 (low-sulfur fuel) mg kg⁻¹-fuel.” is potentially confusing. The fuel sulfur content is not determined by an emission measurement (unless this study did something unique and different), but by an analysis of the fuel. So, calling it an Emission Index is confusing. (And an emission measurement would not be of S but of SO₂ + SO₃ +H₂SO₄ and converting each

of those to represent the equivalent mass of S). Common parlance is to use FSC and units are commonly ppm (parts per million by mass). ppm is equivalent to mg/(kg-fuel), but ppm makes it more clear that the measurement is by fuel analysis, not by an emission measurement. And ppm is used in the caption to Figure 2a, so it would be better to be consistent.

Page 7, line 27: "regimesand" needs a space to read "regimes and".

Page 8, lines 17-20: This discussion needs to be revised unless the fuel composition and temperature correlations discussed above in relation to Figure 2 can be fully demonstrated and supported in a future version of the paper.

References: I am a little confused by having two separate lists of references. Maybe this is a specific requirement for Nature submissions. But even if this is what is needed for Nature, the authors need to carefully check the references. Some references (e.g. Epstein, 2014, Schumann 1996, etc.) are cited in the second section of the paper but are included in the first reference list.

Page 14, line 26: Author contributions "...MW applied ...". There does not appear to be an MW author in the list. Maybe "MV"?

Page 16: While I do not think that the figures can stand as they are, if they are included after major rework, I think it would be valuable to indicate in the captions that in 2b and 2c the change in S (195 vs 3) has little effect on the results. Except for the case with noOM in Figure 2b, the dashed and solid curves overlay each other so closely, it is easy to miss that each color of curve actually represents an overlying solid and dashed curve. This is so pronounced that, for both figures, it would be worthwhile to explicitly (and emphatically?) state that the MoMie model results do not show any significant dependence on FSC unless there is no organic contribution, both in the caption and in the discussion of the figures in the main body of the paper. Also, this seems to be at odds with the results from ACM shown in Figure 2a, which used 25 mg/(kg-fuel) rather than 5 or 50 used by MoMie, but which show a significant dependence (factor of ~30) on FSC for 3/41/195 ppm.

Page 17, line 2: "... sensitivity ti the..." probably means "... sensitivity to the..."

Page 18, line 2: "... a Foto of a ..." maybe "... a photo of a ..." or "... a photograph of a ..."

Page 26, line 5: "... conducted approximately homogeneously." Probably means "... conducted approximately simultaneously."

Referee #2

(Remarks to the Author)

A. Summary of the key results

The authors performed unique measurements of particulate emissions (non-volatile and total) and young contrails at cruising altitudes behind a current state-of-the-art single aisle airliner with lean-burn engines CFM LEAP-1A. This aircraft/engine combination is representative of the short-haul fleet for the next ~20 years. The lean-burn combustor (trade name TAPS – twin annular premixed swirler; three generations have been certified) has been utilized commercially in engines developed by GE Aerospace and their market share has been increasing. These engines are known for ultra-low nvPM emissions when operated in the lean mode (on the ground, this mode is engaged above ~30% take-off thrust). The question is whether these clean-burning engines, coupled with clean fuels, could be the answer to reducing contrail impacts with engine technology and increasing share of sustainable aviation fuels.

Another unique aspect of this study was the forced rich-burn operation of the engine, simulating a "dirty" engine with high soot emissions (higher than conventional rich-quench-lean engines tested previously using the same setup), which enabled direct comparison of different fuels, combustor technologies, and ambient conditions using one platform.

The main empirical result is that concentration of ice particles (or the apparent emission index of ice particles) correlated with the total aerosol concentration, regardless of fuel type and the combustor mode. The lean-burn mode with the cleanest fuel with zero aromatics, highest hydrogen content, and near-zero sulfur content, produced ice crystal concentrations indistinguishable from the rich-burn mode burning conventional Jet A-1 fuel (with sulfur content at ~50% of typical commercial fuel).

The authors performed a sensitivity study, extending two existing microphysics models, where they investigated the effects of fuel sulfur content and aromatics (using data for fuels used in the study), low volatility organic aerosol (simulating lubrication oil vapor) and ambient temperature on ice crystal concentrations as a function of non-volatile PM number. Ice particle number in the soot-poor regime (nvPM number EI < 1e14/kg-fuel) is predicted to be very sensitive to all these parameters. The model data in this regime overlap with the pooled empirical data for all fuels tested.

B. Originality and significance: if not novel, please include reference

This study is original and highly significant in the ongoing discussion and policy development around climate impact of aviation non-CO2 emissions. Contrails have been estimated to have a global warming potential of the same order of

magnitude as the cumulative CO₂ emissions from aviation. This study can shape the direction of future research and regulatory development, focusing on precursors of secondary aerosols that can nucleate ice at cruising altitudes. This study also puts the existing contrail models into question, especially concerning the application for lean-burn engines and the effects of fuel sulfur content and organic material in the plume.

C. Data & methodology: validity of approach, quality of data, quality of presentation

The authors utilized established methods well documented in their previous work using the same experimental platform. The data analysis steps, and treatment of outliers are well documented in Methods. The figures are of high quality, accompanied by appealing photographs from the test campaign.

D. Appropriate use of statistics and treatment of uncertainties

The methods used are appropriate for the large and complex datasets. Measurement uncertainties for the key variables (particle counting, emission indices, ice particle counting) are listed with references to authors' previous work that dealt with them in detail.

E. Conclusions: robustness, validity, reliability

The conclusions based on the empirical results are robust, based on state-of-the-art measurements using well-documented methods. The paper signals a necessary shift in contrail impact modeling, especially for low-soot engine technologies, and regulatory measurements. These findings may be useful for potential developments of hydrogen combustion engines using a similar architecture with lubrication oil vapor vented into the hot exhaust stream.

F. Suggested improvements: experiments, data for possible revision

1. Interpretation of the fuel effects:

In Figure 1, the authors show the EIs of non-volatile and total PM number for the reference Jet A-1 fuel. In Figures 2 and 3, data for all fuels are pooled into common symbols. In the introduction, the authors recall that lower-aromatic SAF reduces soot emissions (and corresponding ice particle concentration), citing earlier NASA and DLR campaigns, yet they do not compare their data across the five fuels for both rich-burn and lean-burn modes. My conclusion from Figures 2 and 3 is that the fuel composition effects were not discernible, especially in the soot-poor regime, with high variability due to ambient conditions and the effects of volatile PM.

It would be useful to discuss the effects of fuel composition (or the lack of it) in the empirical data. The fuel composition effects are clearly described only in the model sensitivity study. The lack of discussion about fuel composition confused me after reading the statement in conclusions (page 9, paragraph 2):

"More experiments are required to investigate contrail formation with sulfur-free fuel in order to test whether sulfur-free fuels could reduce contrail ice crystal numbers in the lean-burn mode. First indications were found for RQL combustors, that low-sulfur fuels might reduce soot particle activation and thereby contrail ice crystals (Wong and Miake-Lye, 2010; Voigt et al., 2021; Jones and Miake-Lye, 2023; Märkl et al., 2024; Yu et al., 2024)."

This study used an ultra-low (3 ppm) sulfur fuel with zero aromatics, which I believe is the lowest sulfur content fuel tested using this platform. The authors' previous work with 100% HEFA SAF reports 7 ppm sulfur (<https://doi.org/10.5194/acp-24-3813-2024>). 3 ppm is extremely challenging for research (contamination) and impossible for commercial applications. Was there no discernible benefit of the 3 ppm sulfur fuel vs the 195 ppm in the lean-burn regime?

2. Matching of engine conditions between far field and near field sampling:

The engine conditions were matched by combustor inlet temperature (T₃). The aircraft flew at different speeds during the near field measurements: Mach 0.59 vs 0.64-0.8. How did other engine operating conditions relevant for soot emissions compare (p₃, AFR, fuel flow)? Was the slower speed achieved by a different aerodynamic configuration of the aircraft?

3. Fuel analysis methods in Table 1:

Can you please add the ASTM method used for the hydrogen content analysis in your study? There are several methods with different levels of precision and accuracy.

If possible, please include the methods used for the sulfur, aromatics and naphthalene as well.

4. Lube oil vapor data in the ACM model (page 28):

The authors assume lube oil vapor emission index of 25 mg/kg fuel as input for ACM model. What is the basis for this assumption? If we assume that oil vapor emission equal to oil consumption, this value is rather low. Engines with similar architecture (CFM56) consume ~0.1-0.3 kg oil per hour of operation (0.5 kg/hour indicates a maintenance problem and necessary intervention). If we assume 0.1 kg/hr, this corresponds to ~80-100 mg/kg-fuel for the LEAP-1A at nominal cruise. Decker et al. recently reported measured oil consumption values of a CFM56 engine measured on the ground, with ~100 mg/kg-fuel at fuel flow corresponding to cruise. (<https://pubs.acs.org/doi/10.1021/acsestair.4c00184>) Would this higher value significantly affect your results in Figure 2a?

5.nvPM distribution representation in the MoMiE model (page 29):

"In all simulations, soot particles are represented by a log-normal distribution with 35 nm median diameter and a standard deviation of 1.6"

Recent work has reported larger geometric standard deviations of soot / nvPM emitted by a variety of rich-burn engines with a mean value ~ 2 (<https://pubs.acs.org/doi/10.1021/acs.est.4c02538>). What is the sensitivity of your simulation to GSD?

Is the assumption of a log-normal distribution for the lean-burn engine reasonable? Ground measurements behind the LEAP-1B have shown aerosol concentrations at the level of ambient background, essentially a random distribution without any discernible peak (presentation by Rich Moore, NASA: <https://www.aerosociety.com/media/26055/iata-d2-richard-moore-iata-raes-contrails-workshop-10-11-march-2025.pdf>).

5.organic material in the MoMiE model (page 29):

Does OM2 represent lubrication oil vapor? What is the difference between OM2 and LOM in ACM in terms of ice nucleation ability?

G. References: appropriate credit to previous work?

The authors reference previous work appropriately.

H. Clarity and context: lucidity of abstract/summary, appropriateness of abstract, introduction and conclusions

The abstract reads well and can attract a wider audience. One small suggestion for the first sentence: "Contrail cirrus are a major contributor..." -> "Contrail cirrus clouds are a major contributor..." or "Contrail cirrus is..." I am not aware of the word cirrus being the same in singular and plural form.

The introduction and conclusions are equally clear, they well summarize the current state of the art and the gap this study addresses.

Page 9, Line 11-13: "Also, engine-to-engine variability and effects of engine deterioration on emissions need to be explored to assess emissions at cruise." While this is certainly true, I found this sentence out of place here; it is not a conclusion or recommendation based on the results of this study and previous sentence. I think it could be deleted without any loss of information.

Referee #3

(Remarks to the Author)

The key results in the paper refer to ice crystal number concentrations (ICNC) in contrails do not decrease with reducing soot number emissions. i.e. despite the change in fuel content and lean engine technologies the production of organic volatile particles provide surface area of ice crystals to condense within the contrail. The data shown here is based on correlations and not explicitly evidenced by looking at residuals of ice crystals to identify organics within the ice crystal.

The manuscript is well written but limited data analysis and uncertainty analysis is presented (see more below).

Also, I believe that when water vapour mixing ratios and temperature is taken into account, one would find that the RH_{ice} are so high in the contrails, that it is sufficient just to have small particles of any composition, for water/ice to condense/freeze in a contrail. The properties of the aerosol become secondary at such high water vapour emissions and low T. It is therefore clearly extendable for thermodynamic properties that the only way a reduction in contrail ICNC can be achieved is that if total particle number concentrations are reduced. The fact that a reduction in soot does not produce a reduction in ICNC is not to do with the properties of the soot aerosol, but with the presence of total particle number concentration in the plume. Water vapour mixing ratios were measured, I would have liked to see the calculations to show what the RH_w or RH_i is in the plume to show that this is well above water saturation and thus given any type of particle present (organic, soot or otherwise) condensation and subsequent freezing is warranted due to the high supersaturations in the contrail plume and low temperatures below -40 C. The RH_i ranges are given in Tables 2 and 3, but a calculation to show that the RH_w is very high in the plume would be more effective and a clear validation of why one would expect ice to form on the organic volatile fraction as well. In this regard a study by Ponsoby et al. (2024, <https://doi.org/10.5194/acp-24-2045-2024>) should be cited who showed contrail ice can form on jet lubrication oil that is discussed in this work.

The results here are in this regard not as novel but the measurements certainly are. These measurements are hard to do and I commend the author team on how they planned the measurements. The burning of different fuel types, modifications of the Airbus Engine to burn different fuel type, the coupling of the two air craft, the near and far field measurements in and out of the plume and flying directly in the plume of an aircraft all take a tremendous amount of planning and coordination and engagement with stakeholders. The results themselves are not surprising and are predictable from previous studies on ice nucleation onto organics and organic and sulfate coated soot particles. Numerous studies have shown that around RH_w = 90% at temperatures below -40C organic particles of small sizes and soot particles form ice, so in a contrail that has

RHw>100% it is unsurprising that ICNCs form on both organic particles and soot.

The results nicely show that $E_{\text{I total}}$ correlate with ICNC and that $E_{\text{I total}} > E_{\text{I nv}}$. And the conclusion that ICs form on the volatile organic fraction is a reasonable conclusion based on correlation but is not explicit evidence. As such this should be clarified in the manuscript for example in the abstract, the use of the word evidence is not appropriate because no direct evidence is provided that the ice crystals formed on the organic components or non volatile components, this is based on the good correlation between $E_{\text{I total}}$ and ICNCs.

The size distributions of the ice crystals could be shown because it seems from the methods that these would be available

More information on the uncertainties and how they are derived and why the uncertainty for $E_{\text{I total}}$ are different than for $E_{\text{I nonvolatile}}$ would be important to include in the methods section.

Can IC shattering be excluded in the measurements of ice crystals, if so why and can these be presented in the manuscript or the methods section? Can an uncertainty in the IC size distribution and number densities be provided?

Figure 2 is shown for the non volatile particle number (on the x-axis), can this be shown also for the total particle number as well.

In the introduction the authors mention that the CO₂ and non-CO₂ warming is similar, based on these new engines studied here, can a statement with regard to the two different warming components be made. Is the contrail warming considered still to be similar to the CO₂ warming, more or less? Since NO_x and CO were also measured.

There are some typos and additional points that should be carefully checked:

Page 1: What is the effect/role of increased water vapor emission in the contrail plume? This should be addressed in the background and discussion parts.

Page 3 Line 12: Heymsfield et al. 2010 is cited here but it is missing in the reference list for the main text.

Page 5 Line 22 and Line 24: Please give a number for the lower and higher $E_{\text{I nv}}$

Page 6 Line 2: It seems the authors assume that soot is equivalent to total non-volatile particles. Please provide the rationale for this point. The other possible solid particles, e.g., metal due to friction of engine components, should be mentioned, even though as minor components.

Figure 2: The legend is hard to read, x-axis should be extended to lower values; table 1 is irrelevant to the split point.

Figure 2b, the solid and dashed line should also be indicated in the legend for the ease of readability.

Page 7 Line 8: How much is the maximum water supersaturation ratio? We expect the authors to provide the observations on the atmospheric conditions, e.g., relative humidity and temperature, to further investigate the dependence of ice number on the atmospheric condition, to examine the role of aviation emissions (non-volatile and total) on ice number under prescribed atmospheric regimes (for instance, a lower and a higher relative condition)

Page 7 Line 29: What does ambient aerosols mean? Does it mean atmospheric background aerosols without aviation emissions?

Page 8 Line 1: There is no direct evidence showing soot and volatile particles contribute to ice formation. Can the authors provide ice residual compositions?

Page 8 Line 2: Please refer to which figure or table?

Page 8 Line 7: Suggest ice crystal formation processes rather than ice nucleation theory .

Figure 4: What is the coagulation effect of particles? given the high number concentration particles. It is included in the model (page 29, line 7), but its effects on the observations are not discussed. Is the time scale (0.1s, 1.0s and 10s) based on observation or simulation? Or randomly guess? Does the RQL combustion not have venting of lubrication oil and associated oil drops?

Page 19 Line 6-7: Is there any direct evidence on that soot first forms droplets and then freezes? Ice can also form onto soot directly.

Page 19 Line 11-12: A paper is recommended as an appropriate reference <https://doi.org/10.5194/acp-24-2045-2024>

P17 Line 2: to but not ti, typo

P17 Line 17: It is modelled but not observed.

P18 Line 2: Photo not Foto

Page 22: CO₂ and NO_x are not relevant to the theme of this study and the results are not discussed.

Page 23: H₂O data and RH results are not presented (except the latter in tables).

Page 29: Line 14-15: A reference is needed for the soot particle size distribution used for models and in general the particle size distribution used for the modeling work.

Referee #4

(Remarks to the Author)

I co-reviewed this manuscript with one of the reviewers who provided the listed reports.

Version 2:

Reviewer comments:

Referee #1

(Remarks to the Author)

The authors have done a tremendous job in responding to reviewer comments. The paper has been improved significantly with the updates that have been done. All of my comments (Reviewer 1) have been carefully discussed and responses to each comment have been carefully considered and included in the improved draft manuscript. The paper is much more detailed and nuanced due the improvements that the authors have made.

While I was impressed by the importance of the experimental results in the first draft, the comparison with the model results and the related interpretation was lacking in that original version. Now the data are presented better and in such a way that the model results can be compared to the data. And the associated discussion and interpretation is now detailed and relevant.

This is an important paper and I would strongly encourage its publication.

Referee #2

(Remarks to the Author)

In my review, I have focused on assessing the improvements / changes in the revised manuscript and will thus forego the summary of the paper provided by referees for the original version.

The original submission, while novel, did not provide a convincing explanation of the observations using models. This issue was raised consistently by all referees. The data visualization was unclear: the effects of fuel composition were not visible in the cloud of data points for all fuels tested. Similarly, the effect of ambient temperature in the soot-poor regime was not shown and discussed sufficiently.

The central question in my review is whether the authors now provide a robust and compelling explanation of their observations and claims of ambient temperature and fuel composition effects across the different soot regimes.

The authors write that "Our results reveal the impact of current lean-burn engine configurations, fuel composition, and ambient temperatures on contrail ice crystal numbers. "

In my view, the authors addressed all referees' comments appropriately and significantly improved their analytical approach by narrowing the results down to cases where they can reasonably investigate the effects of fuel sulfur content under similar atmospheric conditions and also examine the effects of ambient conditions across different fuels and combustion regimes. The critical update is the new Figure 2, which is significantly improved compared to the previous version. This separation and update of their model provides more conclusive intercomparisons between the model and measurement.

Unfortunately, it seems that the FSC effect could be investigated only at two FSC levels, without the cleanest fuel at 3 ppm (only model line included in the new Figure 2a). Also, the highest FSC of 192 ppm is still very low, significantly lower than typical Jet A-1 in Europe (~400-800 ppm). It would be interesting to see how the model predicts ice particle numbers for representative FSCs (Europe, North America, etc.).

While the new Figure 2 is significantly improved, it took me a while to decipher the meaning of some of the symbols and color-coding (in some cases, unsuccessfully – see below).

: Specific comments

Figure 2

Figure 2a

- The data points for the rich burn mode are above the 1:1 line and above the ECLIF1-3 data shown in panel 2b. Is it due to a

different size cutoff for nvPM (14 nm mentioned) than used in previous studies (<14 nm fraction can be ~15% of the number size distribution), or the effect of volatile PM? I am aware that the rich burn mode was forced and thus may not operate the same way as an RQL engine and may have significantly higher unburned HC emissions.

Figure 2b

- ECLIF 1-3 data are confusing. I assume the symbol shape and fill color differentiate engine and fuel type? The gray-filled circle is the same as the one for the "Jet A-1 lean" data.

- No color scale for the SPK-L points – I assume dark blue is the lowest temperature? Is white supposed to be the highest temperature?

- x-axis ticks and labels are inconsistent between panels a and b

- Y-axis label is inconsistent between panels a and b – unit: kg fuel vs. kg-fuel

Suggestions for simplification:

- Use one unique shape for all the ECLIF 1-3 data and differentiate Jet A-1 and SAF / or SAF blends with a different fill color

- Use different shapes for SPK-H, SPK-L and Jet A-1 but use the same color map (temperature) for all these points

P1 L26 – "current lean-burn" – an important addition as maintaining low nvPM emissions is crucial; the issue of high ice crystal numbers is linked to fuel sulfur and lubrication oil venting; The latter might require a significant system redesign across all future engine types (a closed loop system).

P8, L6 "volatile organic species form venting of the..." change form to from

P8, L10 "EI_{org} of 5 mg/kg fuel" – while the authors explain the reasoning behind the assumed oil emission index (manufacturer data), the origin of the EI_{org} value is unclear. Further in Methods on Page 36, the authors write that the predicted HC emissions at cruise are 50-66 mg/kg. Was the value of 5 mg/kg assumed as the "activated fraction" to tweak the model to get good agreement with observations?

P11, L28 "...updates are probably worth considering." – FSC reduction in fossil jet fuel has been the topic of active research and regulatory development. Indeed, especially for lean-burn engines, low-sulfur fuel appears to be key to reducing contrail RF.

Table 1

If possible, for quality checks, please add density (at 15 degC) and lower heating value (LHV) with the methods used to the table, if this data is available.

Table 4

EI_{nv} for the forced reach burn: median is higher than the reported measurement range – maybe there is a typo? This was carried over from the original version.

Referee #5

(Remarks to the Author)

The authors present measurements behind the latest generation of aircraft engines burning commonly used and future fuel types. These measurements are absolutely what is needed for a clearer picture for the future of aviation engine design and fuel optimization. They find that even in fuels and engine burn combinations that produce little soot, the ice crystal concentrations produced by the aircraft are similar as to when soot emissions are not actively reduced. This points to the formation of the crystals on other byproducts from the engine combustion e.g. organic vapors and sulfates and from the engine itself, via lubrication oil. These findings confirm previously hypothesized and simulated contrail ice crystal formation processes and as such, are an important addition to our understanding of contrail cirrus formation. That said, the authors often overstate the novelty of their findings rather than presenting the work as a confirmation of what has already been presented from theory and simulations. Nevertheless, due to the broad interest in the climate impacts from aviation and the fact that these measurements clearly show that switching to leaner combustion and newer fuels is not the silver bullet needed to solve the contrail cirrus problem, the manuscript is well suited for publication after some of the novelty is toned down and the following comments are considered.

General comments:

Page 1 Line 24-26: This is not exactly true as several of the cited studies have already tested and simulated the ability of volatile particles to initiate ice crystal formation. This should be toned down/reformulated as it is the first time this has been observed behind an aircraft.

Page 6 Line 18-20: This is higher than the EI_{nv} in the rich burn mode. How was Elice in those conditions and does this mean that the signal from volatile particles is already present even in the rich burn environment since the Elice in lean burn is higher than EI_{nv} in rich burn?

Page 6 line 28-30: What else could be responsible for the emissions? I don't really understand the question here as there is clearly a source of particles that is missing and needed to explain the high concentration of ice crystals observed.

Page 7 line 11-15: It is not immediately clear what this is effectively saying, is this supposed to be motivation for other effects

as with these two examples, the difference in Elice can almost be explained by the difference in sulfur contents (ratio of 3.2 and 2.6, respectively)?

Page 8 line 17-20: The word reveal here is a bit misleading. It has already been anticipated that in the absence of soot particles, oil and volatile vapors would act as sources of particles for ice crystals to nucleate on (e.g. Ponsonby et al, 2024; Yu et al, 2024). I would instead use something like “confirm.”

Page 8 lines 25-28: It is not clear what the reasoning behind the smaller crystals is here? Is it that in the absence of larger soot particles, the SS is able to reach higher levels and thus more of the smaller and homogeneously sized sulfate and volatile particles activate into droplets and ice crystals at the same time?

Page 9 lines 17-20: It should be made clearer if this related to the soot out competing the oil/organics/sulfates for the available vapor or something else going on. Either way, this feature has already been predicted in previous studies (e.g. Yu et al, 2024), so it is not necessarily a new finding as semi presented here. Instead, it would be better to rephrase this as something that seems more likely from this study and a region that should be sampled. However, based on the results presented here, it looks like the best would be to fly at warmer temperatures and with low sulfur fuel and lean engine combustion.

Page 9 line 26-29: This is an overstatement and not a novel finding. It supports an already anticipated result and as such should be rephrased accordingly.

The figure captions could be shortened by including some of the information in the text instead.

Minor comments:

At the beginning of the text there are a few acronyms that could be removed as they are only used once or twice and often make the reading a bit distracting. Some examples are:

Page 2 Line 16: ETS acronym is not used again

Page 2 Line 17: CORSIA

Page 4, Line 21: FADEC is only used twice and therefore could be removed and simplified. That said, the details on the engine injectors seem very specific and not of general importance to the study/ broad readership, especially without a diagram explaining what is meant by these different locations. Consider shortening the discussion here.

Page 3, Line 10: Consider adding fuel after Jet A1.

Page 4 line 26: Is the conventional petroleum-based fossil kerosene mentioned Jet A1? If so, include that here.

Page 4 and 5: It might be helpful for the reader to have it clear that 7 fuels were tested (as shown in Table 1) and have them described a bit more clearly in the text. For example the description of SPK-L and H could be streamlined.

Page 5 Line 26: consider just writing Jet A1 since it has already been made clear that it is fossil-petroleum based.

Page 6 Line 21-22: Stating that these are the first measurements behind a lean engine seems unnecessary here as this has already been established.

Page 8 Line 22-23: Consider adding a few references here as these findings are very consistent with what is expected of contrail cirrus formation

Page 9 line 13: Add citation to Ponsonby et al, 2014 who showed that above water saturation and at temperatures below the homogeneous freezing temperature, lubricating oils can act as ice-nucleating particles.

Figure 2: Consider making the outer line of the symbols the temperature. It is not very evident what the temperature of the measurements is with the current colorfill scheme.

Page 10 line 23: Can add Ponsonby et al, 2024 here as they did show that lubrication oils act as sites for ice nucleation in the lab.

Page 10 line 24-25: Again, Ponsonby et al, 2024 did show this in the lab, it is not clear why the reference is coming in the middle of the sentence where it makes it seem as if the ice nucleation aspect of this statement has not been shown?

Editorial comments:

Page 2 Line 25: see also Voigt (2025) may read better if it is entirely in parentheses.

Page 6 line 1: consider adding “like” or something similar instead of “of”

Page 7 line 3: A -> At

Table 1 lines 8-9: Make it clearer that the numbers in the parentheses represent %v if that's true, if not update either way.

Our replies to the comments by referee 1 are given in blue.

Referee #1 (Remarks to the Author):

General Comments:

This is an important, ground-breaking study that shows strong evidence of contrail formation on volatile particulate matter (vPM). This is an exciting and significant finding, since this demonstrates that, even for aviation engines that emit essentially no soot/non-volatile PM (nvPM), contrails can still form. This has not been shown before and provides important new insight into aviation contrails, which are one of the largest climate impacts from aviation. This should be of great interest to communities that are keeping track of advances in the understanding of anthropogenic climate change and of the contribution of aviation to climate change.

We thank the reviewer for highlighting the importance and novelty of the manuscript.

The paper is generally well-written and explains the vPM-contrail connections well. However, attempts to describe the possible effects of fuel composition and ambient conditions on the contrail properties are lacking. If there is data supporting their claims in these areas, that has not been included in the figures or in other ways. So these claims, while interesting and potentially insightful, are not demonstrated in the data presented in this paper. Thus, the claims in the abstract “Our results demonstrate the impact of lean-burn engine configurations, fuel composition, and ambient conditions on contrail ice crystal numbers.” should only read “Our results demonstrate the impact of lean-burn engine configurations (and resulting changes in nvPM emissions levels) on contrail ice crystal numbers.” based on what is presented. More detailed discussion of why follows.

We thank the reviewer for this excellent comment. We agree, and we have revised the manuscript accordingly to now directly show data on fuel composition and ambient temperatures. Based on the reviewers comments we separate the different effects of fuel composition and ambient conditions and now show in Figure 2a observations and model results for similar ambient conditions to investigate a fuel type effect (see also Table 5) and in Figure 2b the effect of ambient temperatures on contrails.

Figure 2 and correlations between model results and data

Figure 2 plots measured ice particle data versus nvPM number data in several ways. Figure 2a plots those data in comparison to model plots for simulations for different fuel compositions. Yet, the measurement data is not labelled as to what fuel was used for each point. Only the engine configuration is shown by symbol color. While knowing the engine configuration (lean/split/rich) is worthwhile, all points in each vertical grouping is the same engine configuration, so using symbol color is not needed. Each vertical grouping could be labelled by a text box or similar and the data symbols could be instead colored by the fuel type to see if there is a correlation between the data and the model results. As presented, the data could be correlated with fuel type. Or it could be random or it could be anti-correlated. Presumably the authors know what fuel was used for each point, but that is not evident to the reader. So, the plot does not show any correlation between data and model according to fuel type.

We are grateful to the reviewer for this excellent comment. We have revised Figure 2 to now directly show observations and model results at similar ambient conditions in terms of temperature, humidity and altitude to investigate a fuel type effect on contrails and give these data in Table 5. We have significantly revised the model to now explicitly calculate both, the nucleation and condensation of lubrication oil vapors and unburned organic hydrocarbon vapors in order to better explain the observations. We hope that the updated figures and the revised ms provide better hints and better constrain the fuel type effect and the engine mode effect on contrails. We also separate the ambient temperature effect from the engine mode and fuel type effect in Figure 2 b. We think that this leads to a novel, more robust and mechanically compelling insight into contrail formation in the low-soot regime.

Updated Figure 2 a

Figure 2 Correlation between the non-volatile (i.e. soot) particle number emission index (El_{nv}) and the contrail ice emission index (El_{ice}) in lean-burn and forced rich-burn combustor conditions for different fuels.

Figure 2 | a Dependence of contrail ice crystal numbers on combustion mode and fuel

composition; Measured data: Mean non-volatile (El_{nv} ; $d > 14$ nm), and ice (El_{ice} ; $d > 0.6$ μm) particle number emissions indices per kg-fuel measured at similar atmospheric conditions (218K, Table 5) for Jet-A1 (black circles) and blended HEFA fuel (green circles). Cruise measurement data are shown by symbols and model results by lines. The thin dotted gray line is the 1:1 line. Background information on the data and fuel composition are given in Tables 1 to 5.

In the lean-burn combustion mode, high contrail ice crystal numbers were measured despite low non-volatile (i.e. soot) particle numbers. In both, the forced rich-burn and the lean-burn combustion mode, soot and contrail ice particle numbers are reduced for the HEFA blend with lower aromatic and sulfur content than Jet A-1.

Model simulations: The lines show results from the updated aerosol and contrail microphysics model (ACM, Yu et al., 2024) at similar ambient conditions (218 K, 10 K below T_{SA} , Table 5) for lean-burn and rich-burn combustion modes and for two specific fuel composition cases (Jet A-1, 192 ppmm fuel sulfur content (black line) and HEFA blend, 75 ppmm FSC (green line), see Table 1). Due to a leakage between the two tank systems, the original 100% HEFA-SPK was contaminated and blended with Jet A-1 in this case, which changed the FSC of the Jet A-1 by 5 % and led to the HEFA blend. For the present study, the ACM is improved by explicitly simulating the nucleation and condensation of lubrication oil with a lubrication oil emission index El_{oil} of 120 $\text{mg kg}^{-1}\text{-fuel}$, as adjusted to near-field particle emission measurements behind the left and the right engine. The contribution of low volatile organic hydrocarbons from incomplete fuel combustion is also considered with an emission index El_{org} of 5 $\text{mg kg}^{-1}\text{-fuel}$. The model simulates the competition between nucleation and condensation of sulfuric acid, organic and lubrication oil vapors with individual nucleation and condensation rates for each species. More information on the mode is given in the methods. In addition to Jet A-1 and the HEFA blend, a sensitivity run is performed for the 100% HEFA-SPK with 3 ppmm FSC (dash-dotted green line). In the rich-burn mode, slightly lower El_{nv} than El_{ice} might point to an additional contribution of non-volatile particles smaller than 14 nanometres to ice nucleation, below the detection limit of the particle counter, or to an additional activation of volatile aerosols, which is not yet covered by the model.

Similar arguments apply to Figures 2b and 2c as well. The measurement data is only colored by lean/split/rich and there is no way to determine if the model's variation with organic matter (OM) contribution (2b) or temperature (2c) correlates with the measured data or not. Again, the data could be correlated, random, or anti-correlated based on what is presented in the Figures. It seems possible that perhaps the wrong versions of the figures were included with the paper, since the

authors make the statement “In the lean-burn mode, a strong dependence of EI_{ice} on ambient temperature is observed and modelled.” Yet that is not shown in the Figure 2c as presented. If the authors have other versions of these figures that actually demonstrate these correlations, they need to present them to make these claims. I am astounded that, with the long list of coauthors on this paper, the above was not caught in review prior to submission. The statements related to fuel composition (including OM) and temperature are just not supported by what is included in the figures.

We thank the reviewer for this comment. We have revised Figure 2 b to now directly show observations and model results for the same fuel type for different ambient temperatures to investigate a potential temperature effect on contrails. We compare observations and model results at different temperatures to solidify our arguments on the impact of ambient conditions on contrails. We have shifted simulations with the MoMie model (Fig 2c) in the methods, as the ACM model used in Fig 2 a and b has been revised to now include nucleation of lubrication oil and organic vapors from unburned hydrocarbons. We hope that this is in line with the reviewer’s suggestions. We think that this comment helped to significantly improve the ms and increases the clarity of the argumentation. By separating ambient temperature and fuel effects, we can provide more conclusive data-model intercomparisons and the updated model provides novel insights into contrail formation theory at low soot emission levels.

Updated Figure 2 b

Figure 2 | b Temperature dependence: Measured data: Non-volatile (EI_{nv} ; $d > 14$ nm), and ice (EI_{ice} ; $d > 0.6$ μ m) particle number emissions index per kg-fuel, circles show data measured in different atmospheric conditions (210-222 K, Table 3) binned in 1K temperature intervals for Jet-A1 (gray circles), and the low and high aromatic HEFA-SPK blends (blue and cyan circles). In order to fill the gap between rich and lean-burn combustor modes, an additional fuel distribution has been defined for this campaign, where the pilot flow in the combustor was slightly increased and the main flow decreased compared to the normal lean-burn mode, shown here as intermediate split data points for Jet A-1 binned in 1K temperature intervals and colour coded by ambient temperature (gray scale). SPK-L blend contrail data are also shown temperature coded (211, 212 and 221 K) for two contrail encounters at different RH_i. The fuel composition, engine settings and ambient conditions for the far-field contrail and the near-field emission measurements are given in Tables 1 to 5. Also, data from the ECLIF 1 to 3 flight series with rich-burn engines for similar atmospheric conditions are shown (Voigt et al., 2021; Märkl et al., 2024).

We find a large variation in measured contrail ice crystal numbers for varying atmospheric temperature conditions in the lean-burn regime. Contrail ice crystal numbers decrease with increasing temperature, as shown for Jet A-1 in the intermediate fuel split point.

Model simulations: To illustrate a potential temperature effect on contrails, results from the updated aerosol and contrail microphysics model (ACM, Yu et al., 2024) are shown for different

ambient conditions (216 K, 218 K, and 220 K) for lean-burn and rich-burn combustion modes and specific fuel compositions: Jet A-1 (gray lines) and low aromatic SPK-L blend (blue lines).

In the lean-burn combustion mode, the model simulates a decrease in contrail ice crystal numbers for increasing temperatures, in line with the observations. We find a reduction in soot and ice particles for the SPK-L blend compared to Jet A-1. Lower, but still substantial contrail ice crystal numbers for the low-sulfur SPK-L blend point to a novel contrail ice nucleation pathway in the low-soot regime on volatile particles, potentially nucleated from lubrication oil and organic vapors. In the rich-burn combustion mode, the model simulates a decrease in soot and contrail ice particles in agreement with contrail observations for low-aromatic fuels and modern engines (Voigt et al., 2021; Märkl et al., 2024), confirming the classical theory of contrail ice formation mainly on emitted soot particles (Kärcher, 2018).

But, beyond that, all of the positive discussion of comparisons between model and data ignore some obvious short-comings that are not discussed at all. For the “forced rich-burn” cases, the model results all show a very tight clustering at that range of nvPM ($\sim 10^{15}$). Yet the measured data has as wide a vertical spread for this case as for the other two (lean, split). To say that there is any correlation between models and data is very tenuous. Again, perhaps if the measured data were colored by fuel properties (2a, 2b) or temperature (2c), there might be some correlation for the lean and split cases, but basically there is a wide scatter (\sim one order of magnitude) in the vertical direction for all clusters of points.

We thank the reviewer for this comment and have changed Fig. 2 accordingly, see also reply to previous comments.

The most significant shift in the range and centroid of the data clusters is between the two lean-burn clusters (at $\sim 10^{11}$ and $\sim 10^{12}$) where all the models are totally horizontal with no variation. What is the difference between these two lean-burn clusters (different fuels?)? That seems like the most significant variation amongst the data and it not captured at all by the models. Yet this is not even mentioned in the paper.

The reviewer is correct, these are observations of different fuel types, SPKs with low and high aromatics, respectively. We have clarified this and have changed Fig. 2 b accordingly, see also reply to previous comments.

There is also a smaller shift in range and centroid for the two rich-burn cases, but that might be consistent with the slope of the well-known relation between EI_{ice} and EI_{nvPM} in the soot-rich region. But those two rich-burn cases are so close in nvPM that there is probably little new insight from this.

We took out the temperature dependence of the rich-burn data points as suggested and now show and discuss rich-burn data at similar ambient conditions in Fig. 2 a, see above. The fuel composition dependence that has previously been observed is confirmed as shown in Fig. 2 b.

Basically, all these comparisons with the two models’ results add nothing to the paper as presented. The data seems to not depend on whether nvPM is present or not, which is a main conclusion of the paper. That could be shown in a single plot of the data like in Figure 2, but with no model curves included. (i.e., use any of 2a,2b,2c with the model curves removed.) Any conclusions about fuel composition and temperature dependences are not supported by Figure 2 plots. And even if the authors later label the measured data to show some correlation for the lean-burn and/or split cases, they will need to explain why the models disagree so strongly with the data for the forced rich-burn cases, where the model results have no spread and the data has the same spread as for the other cases.

As reply to this comment we shifted the comparison to the MoMie model (Rojo et al., 2015) in the methods and used this model for a sensitivity study on the effects of variations in emission rates of organic vapors in Figure M2. As we included suggestions by the reviewer to separate the fuel and the temperature effects on contrails in Fig 2 a and b, we think that the comparison to the models now

adds to the interpretations of the results. As no measurements of particle compositions of the small volatile particles in flight exist, models are required to support the interpretations and the conclusions. We revised the manuscript accordingly.

Additionally, and importantly, the model used in Figure 2a (ACM) seems to provide very different results than those for the model used in Figure 2b (MoMie). While MoMie uses OM values of 5 and 50 (as well as 0 and 500), ACM uses a LOM of 25 which might be expected to lie between 5 and 50. MoMie would suggest that FSC of 3 and 195 would lie essentially on top of each other, while ACM shows them a factor of 30 different. There is no discussion of this huge discrepancy between the models. And all of these model calculations are a distraction from the main point that contrails can form on vPM. A separate paper on model comparisons to this data might make more sense.

Given that:

- 1) the models disagree with each other
- 2) neither model captures the spread behavior for the forced-rich case
- 3) the data are not plotted to show trends with fuel type/organic level/temperature, the data cannot be compared with model results

It seems that the model results should be removed from the paper unless these issues are addressed in the paper.

As reply to this comment, we shifted MoMie model results in the methods.

In the main text, we now show observations compared to updated ACM model results in Fig. 2 a and b. We feel that the model simulations are required to interpret the contrail ice numbers measured in the low-soot regime. Hence the model's agreement to the observations and the studies for different fuel types and ambient conditions allow to interpret the measurements. The comparison allows to significantly advance our understanding of contrail theory and point to potentially novel ice nucleation pathways on volatile particles. For medium and high sulfur fuels, an agreement between model and observations is achieved by simulating contrail formation on volatile sulfate aerosol. For low sulfur fuels, a novel contrail ice formation pathway potentially on lubrication oil and organic vapors are required to explain contrail ice nucleation in the low-soot regime. The comparison of measurements and model results advances contrail formation theory and leads to robust process-based insights in contrail formation in the low-soot regime, as required for Nature.

We now only use the MoMie model for a sensitivity study on the effects of variations in emission rates of organic vapors on contrail ice numbers in Figure M2, for which this model was designed. In this way, we use the MoMie model to provide additional model support for our main conclusion of the paper, that ice nucleation on volatile particles is required to explain the observed Elice in the low-soot regime.

Low Fuel Sulfur fuel analysis and inclusion of these fuels in plots

The discussion and use of fuel sulfur content (FSC) from Table 1 is confusing. First, while often FSC is reported in %mass, as done here, another common reporting unit is ppm (parts per million by mass) rather than %mass. And when very low FSCs are being explored, ppm is much easier to read rather than dealing with so many preceding zeros. But, more importantly, the table shows entries for 0.5 and 1.9 ppm, while the models and plots only seem to consider the cases 3.2 ppm FSC and higher. Were these very low cases (0.5, 1.9) included in the plots or not? If the low cases are in the plots, why did the models only use 3.2 ppm as the lowest FSC case? And what method was used to measure FSC and what uncertainty was ascribed to these values? One standard FSC method is ASTM D5453, which cannot measure below 1 ppm (LOD 1 ppm, with LOQ ~3 ppm (?)). So how was 0.5 ppm FSC measured and what is the uncertainty? (Actually, what is the uncertainty for all low FSC data?) And which data in the plots correspond to these very low FSC values or why was that data not included?

We thank the reviewer for this important comment and we now more explicitly give information on the fuel sulfur content, its detection method and related uncertainties. We also give the detection methods for the fuel hydrogen, aromatic, naphthalene contents in Table 1. The text has been changed to use ppm for sulfur.

Sulfur was measured by D5453. The method is indeed limited to 1 ppm, and we are aware on the difficulties and challenges in measuring very low fuel sulfur contents. The “real” detection limit for FSC might be higher, depending on the company measuring the FSC and its transport and fuel handling. In order to comply with the information for ASTM D5453, table 1 has been adapted for SPK-H to „<1“. With respect to uncertainties, the following table shows the repeatability and reproducibility of the measurements at low sulfur concentration end.

Repeatability (r) and Reproducibility (R)		
Concentration (ppmm S)	r	R
1	0.2	0.6
5	0.6	1.9

Detailed comments:

Abstract page 1 line 27-28: As stated above, the phrases “fuel composition, and ambient condition” need to be removed unless and until the correlations described in discussion of Figure 2 can be supported by data which might be included in the paper in the future.

We thank the reviewer for this comment; we think that we have addressed the concerns and that we have solidified our arguments by modifying Figure 2 and the text. We have moderated the wording and changed demonstrate and to reveal. We think that the reworded sentence in the abstract is now scientifically correct and is justified by the revised figures and text.

Our results reveal the impact of current lean-burn engine configurations, fuel composition, and ambient temperatures on contrail ice crystal numbers.

Page 7 lines 4-5. While technically not incorrect, the language “fuel sulfur content with EI ranging from 195 (below world average) to 3 (low-sulfur fuel) mg kg⁻¹-fuel.” is potentially confusing. The fuel sulfur content is not determined by an emission measurement (unless this study did something unique and different), but by an analysis of the fuel. So, calling it an Emission Index is confusing. (And an emission measurement would not be of S but of SO₂ + SO₃ +H₂SO₄ and converting each of those to represent the equivalent mass of S). Common parlance is to use FSC and units are commonly ppm (parts per million by mass). ppm is equivalent to mg/(kg-fuel), but ppm makes it more clear that the measurement is by fuel analysis, not by an emission measurement. And ppm is used in the caption to Figure 2a, so it would be better to be consistent.

Thank you for this good explanation, we changed to ppm FSC throughout the ms. Also, we removed the EI in the fuel sulfur content description on P7 lines 4-5.

Page 7, line 27: “regimesand” needs a space to read “regimes and”.

Done

Page 8, lines 17-20: This discussion needs to be revised unless the fuel composition and temperature correlations discussed above in relation to Figure 2 can be fully demonstrated and supported in a future version of the paper.

We have changed Figure 2 and the text accordingly, thank you very much.

References: I am a little confused by having two separate lists of references. Maybe this is a specific requirement for Nature submissions. But even if this is what is needed for Nature, the authors need to carefully check the references. Some references (e.g, Epstein, 2014, Schumann 1996, etc.) are cited in the second section of the paper but are included in the first reference list.

Done

Page 14, line 26: Author contributions “...MW applied ...”. There does not appear to be an MW author in the list. Maybe “MV”?

Done

Page 16: While I do not think that the figures can stand as they are, if they are included after major rework, I think it would be valuable to indicate in the captions that in 2b and 2c the change in S (195 vs 3) has little effect on the results. Except for the case with noOM in Figure 2b, the dashed and solid curves overlay each other so closely, it is easy to miss that each color of curve actually represents an overlying solid and dashed curve. This is so pronounced that, for both figures, it would be worthwhile to explicitly (and emphatically?) state that the MoMie model results do not show any significant dependence on FSC unless there is no organic contribution, both in the caption and in the discussion of the figures in the main body of the paper. Also, this seems to be at odds with the results from ACM shown in Figure 2a, which used 25 mg/(kg-fuel) rather than 5 or 50 used by MoMie, but which show a significant dependence (factor of ~30) on FSC for 3/41/195 ppm.

We are again grateful for this comment. We agree that the FSC dependence on Elice is substantial, as also modeled with the updated ACM model. We changed Fig 2 accordingly and specify the contrail ice crystal number dependence on the fuel sulfur content in the lean-burn mode. In deed, volatile sulfate aerosol may potentially explain the observed contrail ice crystal numbers in the lean burn regime for medium and high sulfur content fuels. At low fuel sulfur contents, lubrication oil and organic vapors are required to explain the observations in the lean burn regime for an SPK with low aromatic content and only 2ppmm FSC.

We still see a benefit from including the MoMie model for a sensitivity study on low volatile organic vapor nucleation and condensation. As this might not be the main dependency, we have shifted this sensitivity and suggest to show this dependence in the methods.

Page 17, line 2: "... sensitivity to the..." probably means "... sensitivity to the..."

This has been changed.

Page 18, line 2: "... a Foto of a ..." maybe "... a photo of a ..." or "... a photograph of a ..."

This has been changed and a second photo has been added to illustrate variable contrail properties in its formation state in the lean burn regime and the potential effect of lubrication oils on contrails.

Page 26, line 5: "... conducted approximately homogeneously." Probably means "... conducted approximately simultaneously." This has been changed.

Our replies to the comments by referee 2 are given in blue.

Referee #2 (Remarks to the Author):

A. Summary of the key results

The authors performed unique measurements of particulate emissions (non-volatile and total) and young contrails at cruising altitudes behind a current state-of-the-art single aisle airliner with lean-burn engines CFM LEAP-1A. This aircraft/engine combination is representative of the short-haul fleet for the next ~20 years. The lean-burn combustor (trade name TAPS – twin annular premixed swirler; three generations have been certified) has been utilized commercially in engines developed by GE Aerospace and their market share has been increasing. These engines are known for ultra-low nvPM emissions when operated in the lean mode (on the ground, this mode is engaged above ~30% take-off thrust). The question is whether these clean-burning engines, coupled with clean fuels, could be the answer to reducing contrail impacts with engine technology and increasing share of sustainable aviation fuels.

Another unique aspect of this study was the forced rich-burn operation of the engine, simulating a “dirty” engine with high soot emissions (higher than conventional rich-quench-lean engines tested previously using the same setup), which enabled direct comparison of different fuels, combustor technologies, and ambient conditions using one platform.

The main empirical result is that concentration of ice particles (or the apparent emission index of ice particles) correlated with the total aerosol concentration, regardless of fuel type and the combustor mode. The lean-burn mode with the cleanest fuel with zero aromatics, highest hydrogen content, and near-zero sulfur content, produced ice crystal concentrations indistinguishable from the rich-burn mode burning conventional Jet A-1 fuel (with sulfur content at ~50% of typical commercial fuel). The authors performed a sensitivity study, extending two existing microphysics models, where they investigated the effects of fuel sulfur content and aromatics (using data for fuels used in the study), low volatility organic aerosol (simulating lubrication oil vapor) and ambient temperature on ice crystal concentrations as a function of non-volatile PM number. Ice particle number in the soot-poor regime (nvPM number EI $< 1e14/kg\text{-fuel}$) is predicted to be very sensitive to all these parameters. The model data in this regime overlap with the pooled empirical data for all fuels tested.

B. Originality and significance: if not novel, please include reference

This study is original and highly significant in the ongoing discussion and policy development around climate impact of aviation non-CO₂ emissions. Contrails have been estimated to have a global warming potential of the same order of magnitude as the cumulative CO₂ emissions from aviation. This study can shape the direction of future research and regulatory development, focusing on precursors of secondary aerosols that can nucleate ice at cruising altitudes. This study also puts the existing contrail models into question, especially concerning the application for lean-burn engines and the effects of fuel sulfur content and organic material in the plume.

C. Data & methodology: validity of approach, quality of data, quality of presentation

The authors utilized established methods well documented in their previous work using the same experimental platform. The data analysis steps, and treatment of outliers are well documented in Methods. The figures are of high quality, accompanied by appealing photographs from the test campaign.

D. Appropriate use of statistics and treatment of uncertainties

The methods used are appropriate for the large and complex datasets. Measurement uncertainties for the key variables (particle counting, emission indices, ice particle counting) are listed with references to authors' previous work that dealt with them in detail.

E. Conclusions: robustness, validity, reliability

The conclusions based on the empirical results are robust, based on state-of-the-art measurements using well-documented methods. The paper signals a necessary shift in contrail impact modeling,

especially for low-soot engine technologies, and regulatory measurements. These findings may be useful for potential developments of hydrogen combustion engines using a similar architecture with lubrication oil vapor vented into the hot exhaust stream.

We thank the reviewer for the appreciation of the challenges and novelty of our work, its importance to pave the way for future climate compatible aviation - and the insightful comments. Answers are included below.

F. Suggested improvements: experiments, data for possible revision

1. Interpretation of the fuel effects:

In Figure 1, the authors show the EIs of non-volatile and total PM number for the reference Jet A-1 fuel. In Figures 2 and 3, data for all fuels are pooled into common symbols. In the introduction, the authors recall that lower-aromatic SAF reduces soot emissions (and corresponding ice particle concentration), citing earlier NASA and DLR campaigns, yet they do not compare their data across the five fuels for both rich-burn and lean-burn modes. My conclusion from Figures 2 and 3 is that the fuel composition effects were not discernible, especially in the soot-poor regime, with high variability due to ambient conditions and the effects of volatile PM.

It would be useful to discuss the effects of fuel composition (or the lack of it) in the empirical data. The fuel composition effects are clearly described only in the model sensitivity study. The lack of discussion about fuel composition confused me after reading the statement in conclusions (page 9, paragraph 2):

“More experiments are required to investigate contrail formation with sulfur-free fuel in order to test whether sulfur-free fuels could reduce contrail ice crystal numbers in the lean-burn mode. First indications were found for RQL combustors, that low-sulfur fuels might reduce soot particle activation and thereby contrail ice crystals (Wong and Miake-Lye, 2010; Voigt et al., 2021; Jones and Miake-Lye, 2023; Märkl et al., 2024; Yu et al., 2024).”

We thank the reviewer for this excellent comment. We revised Figure 2 accordingly and we now separate the dependence of Elice on fuel composition (and combustion mode) from the dependence of Elice on ambient conditions. In particular, we added data from previous ECLIF1 to 3 campaigns to Figure 2, as requested. We think that this additional information makes this manuscript richer and helps to consolidate the findings.

This study used an ultra-low (3 ppm) sulfur fuel with zero aromatics, which I believe is the lowest sulfur content fuel tested using this platform. The authors' previous work with 100% HEFA SAF reports 7 ppm sulfur (<https://doi.org/10.5194/acp-24-3813-2024>). 3 ppm is extremely challenging for research (contamination) and impossible for commercial applications. Was there no discernible benefit of the 3 ppm sulfur fuel vs the 195 ppm in the lean-burn regime?

We thank the reviewer for this important comment and we now more explicitly give information on the fuel sulfur content, its detection method and related uncertainties. The text has been changed to use ppm for sulfur.

Sulfur was measured by D5453. The method is indeed limited to 1 ppm, and we are aware on the difficulties and challenges in measuring very low fuel sulfur contents. The “real” detection limit for FSC might be higher, depending on the company measuring the FSC, transport and fuel handling. In order to comply with the information for ASTM D5453, table 1 has been adapted for SPK+HA to „<1“. With respect to uncertainties, the following table shows the repeatability and reproducibility of the measurements at low concentration end

Repeatability (r) and Reproducibility (R)		
Concentration (ppm S)	r	R
1	0.2	0.6
5	0.6	1.9

In the updated figure 2a for similar ambient conditions, we find a strong dependence of Elice on the fuel sulfur content in the lean-burn regime, see updated Fig 2 below.

Updated Figure 2

Figure 2 Correlation between the non-volatile (i.e. soot) particle number emission index (EI_{nv}) and the contrail ice emission index (EI_{ice}) in lean-burn and forced rich-burn combustor conditions for different fuels.

Figure 2 | a Dependence of contrail ice crystal numbers on combustion mode and fuel

composition; Measured data: Mean non-volatile (EI_{nv} ; $d > 14$ nm), and ice (EI_{ice} ; $d > 0.6$ μ m) particle number emissions indices per kg-fuel measured at similar atmospheric conditions (218K, Table 5) for Jet-A1 (black circles) and blended HEFA fuel (green circles). Cruise measurement data are shown by symbols and model results by lines. The thin dotted gray line is the 1:1 line. Background information on the data and fuel composition are given in Tables 1 to 5.

In the lean-burn combustion mode, high contrail ice crystal numbers were measured despite low non-volatile (i.e. soot) particle numbers. In both, the forced rich-burn and the lean-burn combustion mode, soot and contrail ice particle numbers are reduced for the HEFA blend with lower aromatic and sulfur content than Jet A-1.

Model simulations: The lines show results from the updated aerosol and contrail microphysics model (ACM, Yu et al., 2024) at similar ambient conditions (218 K, 10 K below T_{SA} , Table 5) for lean-burn and rich-burn combustion modes and for two specific fuel composition cases (Jet A-1, 192 ppm fuel sulfur content (black line) and HEFA blend, 75 ppm FSC (green line), see Table 1). Due to a leakage between the two tank systems, the original 100% HEFA-SPK was contaminated and blended with Jet A-1 in this case, which changed the FSC of the Jet A-1 by 5 % and led to the HEFA blend. For the present study, the ACM is improved by explicitly simulating the nucleation and condensation of lubrication oil with a lubrication oil emission index EI_{oil} of 120 mg kg⁻¹-fuel, as adjusted to near-field particle emission measurements behind the left and the right engine. The contribution of low volatile organic hydrocarbons from incomplete fuel combustion is also considered with an emission index EI_{org} of 5 mg kg⁻¹-fuel. The model simulates the competition between nucleation and condensation of sulfuric acid, organic and lubrication oil vapors with individual nucleation and condensation rates for each species. More information on the mode is given in the methods. In addition to Jet A-1 and the HEFA blend, a sensitivity run is performed for the 100% HEFA-SPK with 3 ppm FSC (dash-dotted green line). In the rich-burn mode, slightly lower EI_{nv} than EI_{ice} might point to an additional contribution of non-volatile particles smaller than 14 nanometres to ice nucleation, below the detection limit of the particle counter, or to an additional activation of volatile aerosols, which is not yet covered by the model.

With the updates on nucleation and condensation of sulfuric acid, lubrication oil and organic vapors from incomplete fuel combustion, the model captures the contrail ice crystal numbers measured in lean-burn and rich-burn combustion modes for Jet A-1 and HEFA blend. In the rich-burn combustion mode, contrail ice formation mainly occurs on emitted soot particles. The strong dependence of observed and modelled contrail ice crystal numbers on the fuel sulfur content

suggests a potential contrail ice nucleation pathway on volatile sulfate aerosol in the lean-burn combustion mode.

Figure 2 | b Temperature dependence: Measured data: Non-volatile (EI_{nv} ; $d > 14$ nm), and ice (EI_{ice} ; $d > 0.6$ μm) particle number emissions index per kg-fuel, circles show data measured in different atmospheric conditions (210-222 K, Table 3) binned in 1K temperature intervals for Jet-A1 (gray circles), and the low and high aromatic HEFA-SPK blends (blue and cyan circles). In order to fill the gap between rich and lean-burn combustor modes, an additional fuel distribution has been defined for this campaign, where the pilot flow in the combustor was slightly increased and the main flow decreased compared to the normal lean-burn mode, shown here as intermediate split data points for Jet A-1 binned in 1K temperature intervals and colour coded by ambient temperature (gray scale). SPK-L blend contrail data are also shown temperature coded (211, 212 and 221 K) for two contrail encounters at different RH_i. The fuel composition, engine settings and ambient conditions for the far-field contrail and the near-field emission measurements are given in Tables 1 to 5. Also, data from the ECLIF 1 to 3 flight series with rich-burn engines for similar atmospheric conditions are shown (Voigt et al., 2021; Märkl et al., 2024).

We find a large variation in measured contrail ice crystal numbers for varying atmospheric temperature conditions in the lean-burn regime. Contrail ice crystal numbers decrease with increasing temperature, as shown for Jet A-1 in the intermediate fuel split point.

Model simulations: To illustrate a potential temperature effect on contrails, results from the updated aerosol and contrail microphysics model (ACM, Yu et al., 2024) are shown for different ambient conditions (216 K, 218 K, and 220 K) for lean-burn and rich-burn combustion modes and specific fuel compositions: Jet A-1 (gray lines) and low aromatic SPK-L blend (blue lines).

In the lean-burn combustion mode, the model simulates a decrease in contrail ice crystal numbers for increasing temperatures, in line with the observations. We find a reduction in soot and ice particles for the SPK-L blend compared to Jet A-1. Lower, but still substantial contrail ice crystal numbers for the low-sulfur SPK-L blend point to a novel contrail ice nucleation pathway in the low-soot regime on volatile particles, potentially nucleated from lubrication oil and organic vapors. In the rich-burn combustion mode, the model simulates a decrease in soot and contrail ice particles in agreement with contrail observations for low-aromatic fuels and modern engines (Voigt et al., 2021; Märkl et al., 2024), confirming the classical theory of contrail ice formation mainly on emitted soot particles (Kärcher, 2018).

2. Matching of engine conditions between far field and near field sampling:

The engine conditions were matched by combustor inlet temperature (T₃). The aircraft flew at different speeds during the near field measurements: Mach 0.59 vs 0.64-0.8. How did other engine operating conditions relevant for soot emissions compare (p₃, AFR, fuel flow)? Was the slower speed achieved by a different aerodynamic configuration of the aircraft?

This again is a good comment. P₃ and fuel flow are slightly higher for the far field measurements at higher speed. The impact on EI_{nv} has been simulated based on the MEEM engine model method: In case of identical ambient conditions, the model simulates a difference of less than 10% in EI_{nv} . When taking into account the ambient conditions (colder temperatures for far field measurements) the model simulates around 20% difference between nearfield and far field measurements.

3. Fuel analysis methods in Table 1:

Can you please add the ASTM method used for the hydrogen content analysis in your study? There are several methods with different levels of precision and accuracy.

If possible, please include the methods used for the sulfur, aromatics and naphthalene as well.

We now give the ASTM fuel analysis methods in table 1 and give more information on reproducibility and repeatability of the method for the detection of the fuel sulfur and fuel hydrogen content, as requested by the reviewer.

Hydrogen content was measured by ASTM D3701 with a repeatability of 0.09 % mass and a reproducibility of 0.11 % mass. Sulfur content was measured with ASTM D5453 with a detection limit of 1 ppm, we changed the table accordingly. As low FSC are difficult to measure, we in addition

give information on reproducibility and repeatability in the table caption. The aromatic content was measured with ASTM D6379 for all fuels, as ASTM D2524 did not give reliable results. The naphthalene content was measured with ASTM D1840.

4. Lube oil vapor data in the ACM model (page 28):

The authors assume lube oil vapor emission index of 25 mg/kg fuel as input for ACM model. What is the basis for this assumption? If we assume that oil vapor emission equal to oil consumption, this value is rather low. Engines with similar architecture (CFM56) consume ~0.1-0.3 kg oil per hour of operation (0.5 kg/hour indicates a maintenance problem and necessary intervention). If we assume 0.1 kg/hr, this corresponds to ~80-100 mg/kg-fuel for the LEAP-1A at nominal cruise.

Decker et al. recently reported measured oil consumption values of a CFM56 engine measured on the ground, with ~100 mg/kg-fuel at fuel flow corresponding to cruise.

(<https://pubs.acs.org/doi/10.1021/acsestair.4c00184>)

Would this higher value significantly affect your results in Figure 2a?

We thank the reviewer for this excellent comment. We have updated the ACM model to now explicitly calculate the nucleation and condensation of lubrication oil vapors and of low volatile unburned fuel hydrocarbon species (organics), with different vapor pressures and nucleation rates. We assume two different low volatile components for the larger chain molecules of the lubrication oils.

In the new model setup, we use 120 mg kg⁻¹ fuel as lubrication oil consumption rate of which 6 to 7% nucleate to new particles (corresponding to the ULVOC fraction of lubrication oil reported in Ungeheuer et al., 2022) and the rest condenses on existing particles. We received new information from the engine manufacturer on lubrication oil consumption rates for the 2024 fleet of Leap 1-A engines of 110 to 140 mg kg⁻¹ fuel, in line with the ground measurement from Decker et al., (2024) of 110 mg kg⁻¹ fuel.

With respect to unburned hydrocarbons, which are smaller molecules compared to the lubrication oil vapors and therefore have different vapor pressures and lower nucleation rates than the lubrication oil vapors, in the updated model, we use Elorg of 5 mg kg⁻¹ fuel for unburned hydrocarbons and organic fuel residual vapors. Model results from the manufacturer of El_{HC} of 50 to 66 mg kg⁻¹ fuel are higher and suggest that only a small fraction of the small unburned hydrocarbon molecules nucleates new particles due to their higher vapor pressure.

Model results suggest a stronger effect of the lubrication oil vapors on new particle formation than from the organic and unburned fuel residual vapors.

5. nvPM distribution representation in the MoMiE model (page 29):

“In all simulations, soot particles are represented by a log-normal distribution with 35 nm median diameter and a standard deviation of 1.6”

Recent work has reported larger geometric standard deviations of soot / nvPM emitted by a variety of rich-burn engines with a mean value ~2 (<https://pubs.acs.org/doi/10.1021/acs.est.4c02538>). What is the sensitivity of your simulation to GSD?

This is a good suggestion. The value of 1.6 is used in both models for the presented simulations and the reference to Yu et al. 2024 has been added in the text. We performed a sensitivity simulation with both models using a sigma of 2.05 (instead of 1.6) for soot as suggested by Durdina et al., 2024. This leads to a decrease in Elice by ~13% with ACM and by ~3% with MoMiE at El_{nv} of 1E15/kg-fuel. The delta in Elice is within the variation of measurement data. Note the sigma value can also affect the ratio of measured El_{nv} (d>14 nm) to El_t. We include the citation of Durdina et al. (2024) in the paper.

Durdina, L., Durand, E., Edebeli, J., Spirig, C., Brem, B. T., Elser, M., Siegerist, F., Johnson, M., Sevenco, Y. A. and A. P. Crayford, Characterizing and Predicting nvPM Size Distributions for Aviation Emission Inventories and Environmental Impact, *Environmental Science & Technology*, 58 (24), 10548-10557, doi: 10.1021/acs.est.4c02538, 2024

Is the assumption of a log-normal distribution for the lean-burn engine reasonable? Ground measurements behind the LEAP-1B have shown aerosol concentrations at the level of ambient background, essentially a random distribution without any discernible peak (presentation by Rich Moore, NASA: <https://www.aerosociety.com/media/26055/iata-d2-richard-moore-iata-raes-contrails-workshop-10-11-march-2025.pdf>).

Non-volatile particle concentrations are very low, three orders of magnitude lower than in rich-quench-lean mode. On the ground ambient background aerosol levels are higher compared to ambient aerosol at cruise altitudes. Therefore, ground measurements might not reveal a discernible peak. We measure non-volatile aerosol larger than 14 nm at low ambient background levels and assume a similar size distribution than for the rich-burn mode, where more data exist. Observations at cruise altitude e.g. with an SMPS might be required in order to resolve the particle size distribution of non-volatile particles in the lean burn mode, in order to answer this interesting question. While the width of the size distribution has a small effect on ice crystal numbers in the rich burn mode, we might expect that at the low nvPM concentrations of the lean burn mode, this effect is even smaller.

5.organic material in the MoMiE model (page 29):

Does OM2 represent lubrication oil vapor? What is the difference between OM2 and LOM in ACM in terms of ice nucleation ability?

Yes, OM2 are organic compounds considered insoluble in water and which are able to condense on soot and volatile particles. OM2 can be seen as the fraction of lubrication oil vapors that condense in the ACM model. We changed the MoMie model description accordingly.

G. References: appropriate credit to previous work?

The authors reference previous work appropriately.

H. Clarity and context: lucidity of abstract/summary, appropriateness of abstract, introduction and conclusions

The abstract reads well and can attract a wider audience. One small suggestion for the first sentence: "Contrail cirrus are a major contributor..." -> "Contrail cirrus clouds are a major contributor..." or "Contrail cirrus is..." I am not aware of the word cirrus being the same in singular and plural form.

Thanks, has been changed to contrail cirrus clouds

The introduction and conclusions are equally clear, they well summarize the current state of the art and the gap this study addresses.

Page 9, Line 11-13: "Also, engine-to-engine variability and effects of engine deterioration on emissions need to be explored to assess emissions at cruise." While this is certainly true, I found this sentence out of place here; it is not a conclusion or recommendation based on the results of this study and previous sentence. I think it could be deleted without any loss of information.

Here we want to point to other potential impacts that can affect emissions and to potential future research topics. We have measured strong variations in total particle numbers between the left and the right engine. Their effect can be seen in updated Fig.3 b on a photo at contrail onset conditions in the lean-burn mode, the left engine produces a much stronger contrail than the right engine. While we do not want to go into the detail in the present manuscript, we see this topic as a motivation for future research. We agree that the sentence is misplaced at the end of the outlook, thank you, and we include a rephrased sentence at an appropriate place earlier in the text in the section before.

"Additional experiments with targeted instrumentations on the ground (e.g. Moore et al., 2017b; Decker et al., 2024; Durdina et al., 2024) and in-flight are needed to explore whether oil emissions are a significant driver for contrail ice formation in the low-soot regime at cruise. Further, effects of engine-to-engine variability (see Figure 3b) and of engine deterioration on emissions need to be investigated."

Figure 3 | a Photo of a contrail forming behind the A321neo taken from the research chase aircraft Falcon | b Photo of the contrail in the lean-burn combustion mode at temperatures close to the contrail formation threshold. The larger contrail optical thickness of the left compared to the right engine might point to different lubrication oil release rates of the two engines, as fuels and engine states are similar at the time the photo was taken. | c Correlation between measured total (i.e. volatile and non-volatile) particle emission indices (EI_t) and ice emission indices (EI_{ice}) per kg of burned fuel in forced rich-burn, lean-burn and fuel split point engine conditions for all fuels at ambient conditions.

Our replies to the comments by referee 3 are given in blue.

Referee #3 (Remarks to the Author):

The key results in the paper refer to ice crystal number concentrations (ICNC) in contrails do not decrease with reducing soot number emissions. i.e. despite the change in fuel content and lean engine technologies the production of organic volatile particles provide surface area of ice crystals to condense within the contrail. The data shown here is based on correlations and not explicitly evidenced by looking at residuals of ice crystals to identify organics within the ice crystal. The manuscript is well written but limited data analysis and uncertainty analysis is presented (see more below).

Also, I believe that when water vapour mixing ratios and temperature is taken into account, one would find that the RH_{ice} are so high in the contrails, that it is sufficient just to have small particles of any composition, for water/ice to condense/freeze in a contrail. The properties of the aerosol become secondary at such high water vapour emissions and low T. It is therefore clearly extendable for thermodynamic properties that the only way a reduction in contrail ICNC can be achieved is that if total particle number concentrations are reduced. The fact that a reduction in soot does not produce a reduction in ICNC is not to do with the properties of the soot aerosol, but with the presence of total particle number concentration in the plume. Water vapour mixing ratios were measured, I would have liked to see the calculations to show what the RH_w or RH_i is in the plume to show that this is well above water saturation and thus given any type of particle present (organic, soot or otherwise) condensation and subsequent freezing is warranted due to the high supersaturations in the contrail plume and low temperatures below -40 C. The RH_i ranges are given in Tables 2 and 3, but a calculation to show that the RH_w is very high in the plume would be more effective and a clear validation of why one would expect ice to form on the organic volatile fraction as well. In this regard a study by Ponsoby et al. (2024, <https://doi.org/10.5194/acp-24-2045-2024>) should be cited who showed contrail ice can form on jet lubrication oil that is discussed in this work. The results here are in this regard not as novel but the measurements certainly are.

We agree with the reviewer that information on maximum supersaturation within the plume would be interesting. Still, measurements of RH_w in the minute-old contrail cannot provide an indication on maximum supersaturation in the early plume, as on average the system relaxes to saturation almost instantaneously if particles are present. Further, measurements in the early jet phase at ~ one second take place where particles have formed already and are in highly variable non-equilibrium state. Therefore, the value of max supersaturation cannot be measured in flight. In non-contrail forming conditions, the supersaturation is per se lower and therefore not representative. The same answer is true for RH_i. Calculations of the time evolution of RH_w and RH_i in the plume have been done previously by Kärcher et al., 2015 (RQL, box model), Bier et al., 2022 (trajectory box model) and Zink et al., 2025 (trajectory box model kerosene with oil particles only and hydrogen). Some simulation results of RH_w from Zink et al are given further below.

Our results are in line with theoretical calculations that in the turbulent non-equilibrium conditions in the early plume, high supersaturations may lead to preferential activation of larger soot particles (~35 nm), while at low soot emission levels, a fraction of the smaller volatile particle mode may get activated into water droplets and then nucleate to ice at ice supersaturation, in line with Köhler theory. While Köhler theory is known since decades, today's contrail climate models either prescribe ice crystal numbers or they only include ice nucleation on emitted soot particles in order to calculate the climate effect. Therefore, current contrail climate effect simulations could miss a significant contribution to the contrail climate effect (e.g. Bock & Burkhardt, ACP, 2019; Teoh et al., ACP, 2024)

These measurements are hard to do and I commend the author team on how they planned the measurements. The burning of different fuel types, modifications of the Airbus Engine to burn different fuel type, the coupling of the two aircraft, the near and far field measurements in and out of the plume and flying directly in the plume of an aircraft all take a tremendous amount of planning and coordination and engagement with stakeholders. The results themselves are not surprising and are predictable from previous studies on ice nucleation onto organics and organic and sulfate coated

soot particles. Numerous studies have shown that around $RH_w = 90\%$ at temperatures below -40°C organic particles of small sizes and soot particles form ice, so in a contrail that has $RH_w > 100\%$ it is unsurprising that ICNCs form on both organic particles and soot.

Today's contrail formation theory suggests soot particles as major driver and nuclei for contrail ice nucleation (e.g. Kärcher, Nature Comm., 2018). This has been shown for RQL combustors (e.g. Voigt et al., Comm Earth Env, 2021; Märkl et al., ACP, 2024). Theory also predicts ice nucleation on volatile sulfuric acid aerosol in the low-soot regime (Kärcher, 2018), but previously no measurements existed to confirm this hypothesis. Also, contrail ice nucleation on volatile lubrication oil particles had not been considered before, there were no model simulations including lubrication oil as potential contrail ice nuclei. We think that our study is novel in this aspect.

Certainly, organic material and other non-volatile particles may contribute to contrail formation - as might be expected given the high supersaturations in the early plume, but this had not been measured and modeled yet. Another important aspect is that the competition for vapors between volatile, semi-volatile and non-volatile particles hasn't been observed yet. In particular, contrail ice crystal numbers could previously be represented by nucleation on the larger soot particles only (e.g. Märkl et al., ACP, 2024, Kleine et al., 2019), as the conventional rich-burn or rich-quench-lean engine technologies emit large amounts of soot particles ($E_{\text{Inv}} > 10^{15}$ n/kg-fuel).

For the current study, we for the first time took the chance to measure in-flight emissions from a lean-burn engine, which showed to have extremely low soot number emissions. Lean burn engines came into operation in the last decade and the fraction of lean-burn engines is continuously growing. This allowed for the analysis of nucleation and condensation of low volatile vapors of sulfuric acid, organic species from unburned fuel hydrocarbons and lubrication oil vapors as well as competition of different gases and particle types for the available water vapor required for ice nucleation in non-equilibrium conditions in the cooling plume.

This are the first observations of high E_{Ice} in the low-soot regime. Also, the range of E_{Ice} has been narrowed down by our measurements. The question was which types of volatile particles can be activated into contrail ice. The measurements were used to update a contrail model to now in addition to soot and sulfate aerosol particles simulates contrail ice nucleation on lubrication oil vapors, this had not been simulated in this way before, and allowed to advance theory on contrail formation, with important impacts on climate. Figure 2 has been updated accordingly.

The results nicely show that E_{Total} correlate with ICNC and that $E_{\text{Total}} > E_{\text{Inv}}$. And the conclusion that ICs form on the volatile organic fraction is a reasonable conclusion based on correlation but is not explicit evidence. As such this should be clarified in the manuscript for example in the abstract, the use of the word evidence is not appropriate because no direct evidence is provided that the ice crystals formed on the organic components or non volatile components, this is based on the good correlation between E_{Total} and ICNCs.

We think that this statement is justified as the measured volatile particle number concentrations are of similar magnitude as the number of contrail ice crystals in the lean-burn regime, while three orders of magnitude less soot was detected. We further improved and advanced the ACM contrail model to now explicitly simulate the nucleation and condensation of lubrication oils and unburned hydrocarbon vapors, in addition to volatile sulfate aerosol and soot particle formation. The model can explain the particle and contrail observations for different engine modes and different fuel composition. Therefore, we think that the statement is correct **"We provide first experimental evidence and theoretical explanations for contrail ice activation on volatile aerosol in the absence of soot."** as we do not mention organic aerosol here.

We further tuned down the notion and changed the following sentence in the abstract to **"Our results reveal the impact of lean-burn engine configurations, fuel composition, and ambient temperature on contrail ice crystal numbers."**

The size distributions of the ice crystals could be shown because it seems from the methods that these would be available

We thank the reviewer for this comment and we now include an additional figure with contrail ice particle size distributions in the methods. We find an effect of the lean-burn mode compared to the rich burn mode and effect for the sustainable aviation fuel HEFA-SPK compared to Jet A-1 on contrail ice crystal sizes. We include this in the main text and the methods.

Contrail particle size distributions

Besides the fuel and engine mode dependent changes in contrail ice particle numbers, we investigate variations in contrail particle size distributions (PSDs). To this end, PSDs of single contrail encounters formed on emissions from Jet A-1 and HEFA-SPK fuels are shown in lean-burn and rich-burn combustion mode in Figure M3 for similar ambient measurement conditions given in Table 5.

Figure M3 | Contrail ice crystal particle size distribution (PSD) of contrail encounters in lean-burn (a, c) and rich-burn (b, d) engine modes for Jet A-1 (c, d) and HEFA-SPK (a, b) fuels at similar ambient conditions (Table 5). The color bar shows the ambient temperatures of 217.9 to 218.5 at emission for the contrail encounters. PSDs of number concentrations per logarithmic bin width were normalized to the bin with highest concentration. The values are calculated to illustrate trends in ice crystal size distributions shown as $dN/d\log D$ and do not reflect absolute number concentrations. Mean effective diameters are 4.47 μm for HEFA blend lean-burn mode, 5.10 μm for HEFA blend rich-burn mode, 4.26 μm for Jet A-1 lean-burn mode and 4.58 μm for Jet A-1 rich-burn mode.

The contrail particle size distributions for each fuel type and engine mode exhibit a slight dependence on initial ice crystal numbers due to changes in fuel type and combustion mode. Lower initial ice crystal numbers, e.g. for the HEFA blend, lead to slightly larger ice crystal mean diameters as the water vapor emissions from the engine are evenly distributed to less ice crystals for the HEFA fuel. Mean contrail diameters are also slightly lower in the lean-burn combustion mode compared to the rich-burn mode in line with slightly higher initial ice crystal numbers in the lean-burn mode. A similar trend has been observed by Voigt et al. (2021) for a semisynthetic jet fuel compared to Jet A-1 in a contrail analysis for similar ambient conditions and contrail age.

More information on the uncertainties and how they are derived and why the uncertainty for E_{total} are different than for $E_{\text{nonvolatile}}$ would be important to include in the methods section.

We now include the following information on uncertainties in the methods for aerosol instruments: Uncertainties in particle number concentration measurements are mainly caused by uncertainties of the low-pressure correction functions, which amount to 7–13% at ambient pressures of 250–350 hPa and may vary slightly between the two CPCs. In the emission index uncertainty analysis, additional contributions arise from aerosol and CO_2 background variability, the uncertainty of the CO_2

measurement, and of the hydrogen-to-carbon ratio of the fuel. Inlet effects are negligible owing to the small particle sizes ($<0.1 \mu\text{m}$). Overall, the uncertainty of the particle emission index in the near-field at 300 hPa amounts to about 10%. Further details on particle measurement uncertainties are provided in Dischl et al., 2024.

Can IC shattering be excluded in the measurements of ice crystals, if so why and can these be presented in the manuscript or the methods section? Can an uncertainty in the IC size distribution and number densities be provided?

Measurements were performed in the upper troposphere/lower stratosphere in thin cirrus conditions. Ice crystal shattering is often found at lower altitudes in mixed phase or very thick ice clouds. We also performed a shattering analysis for the cloud probe data and we can exclude ice crystal shattering for our contrail measurements. Variations in contrail ice crystal size distributions are given in new Figure M3.

Figure 2 is shown for the non volatile particle number (on the x-axis), can this be shown also for the total particle number as well.

The dependence of contrail ice crystal numbers on total particle number is shown in Figure 3.

In the introduction the authors mention that the CO₂ and non-CO₂ warming is similar, based on these new engines studied here, can a statement with regard to the two different warming components be made. Is the contrail warming considered still to be similar to the CO₂ warming, more or less? Since NO_x and CO were also measured.

This is a good question, which needs to be investigated using contrail-climate models. We state in the outlook that simulations would be required to assess the climate effect from lean burn engines. Currently the models assume only ice nucleation on non-volatile soot particles. We expect the additional component from volatile particles on contrail ice crystal numbers to increase warming from contrail cirrus, how much needs to be simulated.

There are some typos and additional points that should be carefully checked:

Page 1: What is the effect/role of increased water vapor emission in the contrail plume? This should be addressed in the background and discussion parts.

We now include the emission index of water vapor for Jet A-1 and HEFA-SPK in table 5. EI H₂O is 1.26 kg kg⁻¹-fuel for Jet A-1 and 1.33 kg kg⁻¹-fuel for HEFA-SPK.

Page 3 Line 12: Heymsfield et al. 2010 is cited here but it is missing in the reference list for the main text.

The reference has been added.

Page 5 Line 22 and Line 24: Please give a number for the lower and higher E_{Inv}

We now include this information in the text as follows:

The measured E_{Inv} is lower compared to Jet A-1 data from older engines with higher soot emissions of the IAE engine V2500 engines ($5.0 \times 10^{15} \text{ kg}^{-1}\text{-fuel}$, Voigt et al., 2021). It is in the range of cruise data from a modern Rolls-Royce Trent XWB-84 engine ($1.0 \times 10^{15} \text{ kg}^{-1}\text{-fuel}$, Dischl et al., 2024) and higher than emissions from an older CFM56-2Ca engine ($3.0 \times 10^{14} \text{ kg}^{-1}\text{-fuel}$, Moore et al., 2017) measured at different ambient and engine conditions.

Page 6 Line 2: It seems the authors assume that soot is equivalent to total non-volatile particles. Please provide the rationale for this point. The other possible solid particles, e.g., metal due to friction of engine components, should be mentioned, even though as minor components.

In aviation emissions research, non-volatile particulate matter (nvPM) and soot are indeed commonly used interchangeably, as nvPM in aircraft exhaust is predominantly composed of soot particles from incomplete fuel combustion. This is particularly true for fuel-rich combustion conditions, characterized by significant soot formation. Other potential solid particle types, such as metallic

particles lubrication oil additives or engine wear debris, are recognized, but are generally considered a minor contribution compared to soot. We do not expect lean-burn operation to significantly alter such metallic particle emissions, therefore we assign the reduction in nvPM to a reduction in soot particles at this point. We added a sentence, explaining that we assign the EInv to soot particles, because soot particles are the major component of non-volatile particles in aircraft exhaust.

Figure 2: The legend is hard to read, x-axis should be extended to lower values; table 1 is irrelevant to the split point.

We modified Figure 2 and increased the font size to facilitate readability.

Figure 2b, the solid and dashed line should also be indicated in the legend for the ease of readability.

We modified Figure 2b, and shifted 2c into the methods. We give the legend to ease readability.

Page 7 Line 8: How much is the maximum water supersaturation ratio? We expect the authors to provide the observations on the atmospheric conditions, e.g., relative humidity and temperature, to further investigate the dependence of ice number on the atmospheric condition, to examine the role of aviation emissions (non-volatile and total) on ice number under prescribed atmospheric regimes (for instance, a lower and a higher relative condition).

We thank the reviewer for this suggestion and now include Figure 2b, where we investigate the effect of variations in ambient conditions on contrail ice crystals. We compare measured to modelled contrail ice crystal numbers for different ambient temperatures and find a good agreement with the improved ACM model for the observed temperature range.

Updated Figure 2

Figure 2 Correlation between the non-volatile (i.e. soot) particle number emission index (EI_{nv}) and the contrail ice emission index (EI_{ice}) in lean-burn and forced rich-burn combustor conditions for different fuels.

Figure 2 | a Dependence of contrail ice crystal numbers on combustion mode and fuel composition;

Measured data: Mean non-volatile (EI_{nv} ; $d > 14$ nm), and ice (EI_{ice} ; $d > 0.6$ μ m) particle number emissions indices per kg-fuel measured at similar atmospheric conditions (218K, Table 5) for Jet-A1 (black circles) and blended HEFA fuel (green circles). Cruise measurement data are shown by symbols and model results by lines. The thin dotted gray line is the 1:1 line. Background information on the data and fuel composition are given in Tables 1 to 5.

In the lean-burn combustion mode, high contrail ice crystal numbers were measured despite low non-volatile (i.e. soot) particle numbers. In both, the forced rich-burn and the lean-burn combustion mode, soot and contrail ice particle numbers are reduced for the HEFA blend with lower aromatic and sulfur content than Jet A-1.

Model simulations: The lines show results from the updated aerosol and contrail microphysics model (ACM, Yu et al., 2024) at similar ambient conditions (218 K, 10 K below T_{SA} , Table 5) for lean-burn and

rich-burn combustion modes and for two specific fuel composition cases (Jet A-1, 192 ppm sulfur content (black line) and HEFA blend, 75 ppm FSC (green line), see Table 1). Due to a leakage between the two tank systems, the original 100% HEFA-SPK was contaminated and blended with Jet A-1 in this case, which changed the FSC of the Jet A-1 by 5 % and led to the HEFA blend. For the present study, the ACM is improved by explicitly simulating the nucleation and condensation of lubrication oil with a lubrication oil emission index El_{oil} of $120 \text{ mg kg}^{-1}\text{-fuel}$, as adjusted to near-field particle emission measurements behind the left and the right engine. The contribution of low volatile organic hydrocarbons from incomplete fuel combustion is also considered with an emission index El_{org} of $5 \text{ mg kg}^{-1}\text{-fuel}$. The model simulates the competition between nucleation and condensation of sulfuric acid, organic and lubrication oil vapors with individual nucleation and condensation rates for each species. More information on the mode is given in the methods. In addition to Jet A-1 and the HEFA blend, a sensitivity run is performed for the 100% HEFA-SPK with 3 ppm FSC (dash-dotted green line). In the rich-burn mode, slightly lower El_{nv} than El_{ice} might point to an additional contribution of non-volatile particles smaller than 14 nanometres to ice nucleation, below the detection limit of the particle counter, or to an additional activation of volatile aerosols, which is not yet covered by the model.

With the updates on nucleation and condensation of sulfuric acid, lubrication oil and organic vapors from incomplete fuel combustion, the model captures the contrail ice crystal numbers measured in lean-burn and rich-burn combustion modes for Jet A-1 and HEFA blend. In the rich-burn combustion mode, contrail ice formation mainly occurs on emitted soot particles. The strong dependence of observed and modelled contrail ice crystal numbers on the fuel sulfur content suggests a potential contrail ice nucleation pathway on volatile sulfate aerosol in the lean-burn combustion mode.

Figure 2 | b Temperature dependence: Measured data: Non-volatile (El_{nv} ; $d > 14 \text{ nm}$), and ice (El_{ice} ; $d > 0.6 \mu\text{m}$) particle number emissions index per kg-fuel, circles show data measured in different atmospheric conditions (210-222 K, Table 3) binned in 1K temperature intervals for Jet-A1 (gray circles), and the low and high aromatic HEFA-SPK blends (blue and cyan circles). In order to fill the gap between rich and lean-burn combustor modes, an additional fuel distribution has been defined for this campaign, where the pilot flow in the combustor was slightly increased and the main flow decreased compared to the normal lean-burn mode, shown here as intermediate split data points for Jet A-1 binned in 1K temperature intervals and colour coded by ambient temperature (gray scale). SPK-L blend contrail data are also shown temperature coded (211, 212 and 221 K) for two contrail encounters at different RH. The fuel composition, engine settings and ambient conditions for the far-field contrail and the near-field emission measurements are given in Tables 1 to 5. Also, data from the ECLIF 1 to 3 flight series with rich-burn engines for similar atmospheric conditions are shown (Voigt et al., 2021; Märkl et al., 2024).

We find a large variation in measured contrail ice crystal numbers for varying atmospheric temperature conditions in the lean-burn regime. Contrail ice crystal numbers decrease with increasing temperature, as shown for Jet A-1 in the intermediate fuel split point.

Model simulations: To illustrate a potential temperature effect on contrails, results from the updated aerosol and contrail microphysics model (ACM, Yu et al., 2024) are shown for different ambient conditions (216 K, 218 K, and 220 K) for lean-burn and rich-burn combustion modes and specific fuel compositions: Jet A-1 (gray lines) and low aromatic SPK-L blend (blue lines).

In the lean-burn combustion mode, the model simulates a decrease in contrail ice crystal numbers for increasing temperatures, in line with the observations. We find a reduction in soot and ice particles for the SPK-L blend compared to Jet A-1. Lower, but still substantial contrail ice crystal numbers for the low-sulfur SPK-L blend point to a novel contrail ice nucleation pathway in the low-soot regime on volatile particles, potentially nucleated from lubrication oil and organic vapors. In the rich-burn combustion mode, the model simulates a decrease in soot and contrail ice particles in agreement with contrail observations for low-aromatic fuels and modern engines (Voigt et al., 2021; Märkl et al., 2024), confirming the classical theory of contrail ice formation mainly on emitted soot particles (Kärcher, 2018).

With respect to maximum supersaturation, Zink et al., ACP, 2025, have provided trajectory-based box model simulations on the time scales of supersaturation with respect to water in the plume for hydrogen and kerosene combustion. Here oil particles of 5 nm radius were the only nucleation particles introduced into the plume (low-soot regime).

Panel a shows the RHw in the plume if microphysics is disregarded and if its simulated. As shown in Bier et al. (2022), the processes happen within the first second of the plume and are inhomogeneously distributed over the plume cross section. Therefore, there is not a single value that can be mentioned. Supersaturation starts at the edge and propagates into the plume.

Page 7 Line 29: What does ambient aerosols mean? Does it mean atmospheric background aerosols without aviation emissions?

Yes, this means ambient aerosols that were measured outside of the plume or atmospheric background aerosol outside of the plume, we added background to be more precise.

Page 8 Line 1: There is no direct evidence showing soot and volatile particles contribute to ice formation. Can the authors provide ice residual compositions?

Ice residuals and their compositions were not measured as the Falcon research aircraft is limited in payload. In future campaigns, one could consider which instrument should be replaced to include ice residual instrumentation. It is to be noted, that ice residuals give indirect information on ice nuclei. Currently there is no instrument to measure the composition of the small nanometer sized volatile particles in contrail conditions. In future the composition of larger volatile plume particles could eventually be measured with filter samples followed by liquid chromatography and mass spectrometry. E.g. Breuninger et al. (2025) have found hints for pentaerythritol ester (PEE, C₂₉H₅₂O₈) in aviation influenced air masses above Germany, and PEE has been identified as a component of aircraft lubrication oil (Ungeheuer et al., 2022).

Breuninger et al., Organic aerosols mixing across the tropopause and its implication for anthropogenic pollution of the UTLS, EGU sphere [preprint], <https://doi.org/10.5194/egusphere-2025-3129>, 2025.

Page 8 Line 2: Please refer to which figure or table?

We refer to Figure 3, as given in the text.

Page 8 Line 7: Suggest ice crystal formation processes rather than ice nucleation theory.
 This has been changed.

Figure 4: What is the coagulation effect of particles? given the high number concentration particles. It is included in the model (page 29, line 7), but its effects on the observations are not discussed. Is the time scale (0.1s, 1.0s and 10s) based on observation or simulation? Or randomly guess? Does the RQL combustion not have venting of lubrication oil and associated oil drops?

The formation of a few very large oil droplets has been reported from ground emission measurements behind RQL engines with an oil venting system in the colder bypass flow (Linke-Deisinger, 2008; Moore et al., 2015). Decker et al. (2024) also find lubrication oil particles in the exhaust of RQL engines, in addition to soot. In contrast, in lean burn engines the venting of lubrication oil into hot areas of the core exhaust can lead to the formation of oil vapors, which may then recondense on existing aerosol or may nucleate new particles. Both processes can enhance contrail ice particle concentrations.

The timescales in Fig. 4 are indicative, based on Kärcher (Nature Comm., 2018), therefore we added // to indicate the range.

Page 19 Line 6-7: Is there any direct evidence on that soot first forms droplets and then freezes? Ice can also form onto soot directly.

The plume temperatures at droplet formation are too high to initiate homogeneous freezing. For typical engine parameters and atmospheric conditions, supersaturation with respect to water is reached above typical ice freezing temperatures, freezing then is terminated upon further cooling at temperatures below 232 K.

As reference, we provide Figure 2 from Kärcher et al., JGR, 2015.

(a) Supercooled water vapor saturation pressure (dashed curve) and average jet plume mixing line (slope $G = 1.64 \text{ PaK}^{-1}$ at $P = 250 \text{ hPa}$) at the formation threshold temperature for a water-saturated atmosphere ($\Theta_G = 231.2 \text{ K}$, thick solid line) and below the formation threshold for $T_a = 220 \text{ K}$ in an ice-saturated atmosphere ($\Theta = 224.6 \text{ K}$, thin solid line) versus plume temperature. The two dotted lines mark shifts of the mixing lines when accounting for uncertainties of humidity and temperature in airborne measurements.

(b) Plume water saturation ratio (solid curves) evaluated along the mixing lines. In the "below-threshold" case, imposed experimental uncertainties in temperature and humidity measurements result in variations in the peak S_{mw} values (dotted curves) of $\pm 10\%$. The plume becomes water saturated ($S_{mw} = 1$) for the first time at $T_x = 239 \text{ K}$, and the peak saturation ratio is reached at $T_\Lambda = 227.4 \text{ K}$. The corresponding results for the threshold case are $\pm 7\%$ and $T_x = T_\Lambda \approx \Theta_G$. Homogeneous

freezing temperatures of contrail water droplets range between 229 and 232K (gray region). All curves neglect water condensation on plume particles.

Page 19 Line 11-12: A paper is recommended as an appropriate reference

<https://doi.org/10.5194/acp-24-2045-2024>

The paper by Ponsonby et al. (2024), is already cited in the main text. We refrain from referencing it in the figure caption 4, as this figure is a sketch of the potential ice nucleation processes and not directly related to Ponsonby et al. (2024).

P17 Line 2: to but not ti, typo

Thanks, has been changed

P17 Line 17: It is modelled but not observed.

Thanks, has been changed to modelled

P18 Line 2: Photo not Foto

Thanks, has been changed

Page 22: CO₂ and NO_x are not relevant to the theme of this study and the results are not discussed.

CO₂ and reactive nitrogen species NO_y are shown in the timeline in Figure M1. In addition, CO₂ is used to calculate emission indices in order to be independent on plume age and measurement position in the plume. We therefore prefer to keep the instrument description in the methods section.

Page 23: H₂O data and RH results are not presented (except the latter in tables).

Base on the reviewer comment, H₂O and RHI data are now included in Figure M1 as given below.

Page 29: Line 14-15: A reference is needed for the soot particle size distribution used for models and in general the particle size distribution used for the modeling work.

The soot particle size distribution is given in Yu et al. (2024) and the citation has been added.

Referee #4 (Remarks to the Author):

I co-reviewed this manuscript with one of the reviewers who provided the listed reports.
This email has been sent through the Springer Nature Manuscript Tracking System NY-610A-SN&MTS

Confidentiality Statement:

This e-mail is confidential and subject to copyright. Any unauthorised use or disclosure of its contents is prohibited. If you have received this email in error please notify our Manuscript Tracking System Helpdesk team at <http://platformsupport.nature.com>.

Details of the confidentiality and pre-publicity policy may be found here

<http://www.nature.com/authors/policies/confidentiality.html>

Privacy Policy | Update Profile

Oberpfaffenhofen, 26 Jan 2026

Referee #1 (Remarks to the Author):

The authors have done a tremendous job in responding to reviewer comments. The paper has been improved significantly with the updates that have been done. All of my comments (Reviewer 1) have been carefully discussed and responses to each comment have been carefully considered and included in the improved draft manuscript. The paper is much more detailed and nuanced due the improvements that the authors have made.

While I was impressed by the importance of the experimental results in the first draft, the comparison with the model results and the related interpretation was lacking in that original version. Now the data are presented better and in such a way that the model results can be compared to the data. And the associated discussion and interpretation is now detailed and relevant.

This is an important paper and I would strongly encourage its publication.

We sincerely thank the reviewer for the insightful comments, which have greatly contributed to enhancing the quality of our manuscript, refining the discussion, and strengthening its overall rigor. We also acknowledge the reference to the additional work by the authors.

Referee #2 (Remarks to the Author):

In my review, I have focused on assessing the improvements / changes in the revised manuscript and will thus forego the summary of the paper provided by referees for the original version. The original submission, while novel, did not provide a convincing explanation of the observations using models. This issue was raised consistently by all referees. The data visualization was unclear: the effects of fuel composition were not visible in the cloud of data points for all fuels tested. Similarly, the effect of ambient temperature in the soot-poor regime was not shown and discussed sufficiently. The central question in my review is whether the authors now provide a robust and compelling explanation of their observations and claims of ambient temperature and fuel composition effects across the different soot regimes. The authors write that “Our results reveal the impact of current lean-burn engine configurations, fuel composition, and ambient temperatures on contrail ice crystal numbers.” In my view, the authors addressed all referees' comments appropriately and significantly improved their analytical approach by narrowing the results down to cases where they can reasonably investigate the effects of fuel sulfur content under similar atmospheric conditions and also examine the effects of ambient conditions across different fuels and combustion regimes. The critical update is the new Figure 2, which is significantly improved compared to the previous version. This separation and update of their model provides more conclusive intercomparisons between the model and measurement. Unfortunately, it seems that the FSC effect could be investigated only at two FSC levels, without the cleanest fuel at 3 ppm (only model line included in the new Figure 2a). Also, the highest FSC of 192 ppm is still very low, significantly lower than typical Jet A-1 in Europe (~400-800 ppm). It would be interesting to see how the model predicts ice particle numbers for representative FSCs (Europe, North America, etc.).

→ We thank the reviewer for the valuable feedback, which has contributed to significantly improving the manuscript and strengthening its robustness. In response, we have now included separated Fig 2a and b into Fig 2 and Fig 3, which compares our observations using an ultra-low sulfur SPK-L fuel to model results assuming a fuel sulfur content of 3 ppm. We hope this addition addresses the reviewer's concerns. While we agree that exploring higher sulfur content fuels would be valuable for future studies, we have opted not to include further simulations at this stage in order to maintain the clarity and conciseness required for the manuscript's current scope.

While the new Figure 2 is significantly improved, it took me a while to decipher the meaning of some of the symbols and color-coding (in some cases, unsuccessfully – see below) : Specific comments
Figure 2 Figure 2a - The data points for the rich burn mode are above the 1:1 line and above the ECLIF1-3 data shown in panel 2b. Is it due to a different size cutoff for nvPM (14 nm mentioned) than used in previous studies (<14 nm fraction can be ~15% of the number size distribution), or the effect of volatile PM? I am aware that the rich burn mode was forced and thus may not operate the same way as an RQL engine and may have significantly higher unburned HC emissions.

→ We acknowledge that the ice data for the forced rich-burn mode lie above the 1:1 line, and we have clarified the underlying reasons in the Methods section and the model description. One contributing factor may be the influence of volatile particles, which can be activated even in the high-soot regime (as observed, for example, by Dischl et al., 2025) and potentially are not adequately represented in the model. A second factor relates to the comparison in Figure 2 and 3. The ice crystal number in contrails is plotted against near-field particle emission data measured at plume ages of < 2 s under similar engine operating conditions. In contrast, the ECLIF1–3 contrail ice data presented in Voigt et al. (2021) and Märkl et al. (2024), as shown in Figure 3, are plotted against simultaneous

far-field non-volatile particle observations in contrails. This difference in measurement context can lead to an apparent shift in the non-volatile particle number emission index (along the x-axis), since complex plume and contrail processing over the first ~2 minutes may promote the growth of non-volatile particles into the detectable size range of our instrument (> 14 nm).

Figure 2b - ECLIF 1-3 data are confusing. I assume the symbol shape and fill color differentiate engine and fuel type? The gray-filled circle is the same as the one for the "Jet A-1 lean" data. - No color scale for the SPK-L points – I assume dark blue is the lowest temperature? Is white supposed to be the highest temperature? → Colour has been changed to dark green for ECLIF3. We removed the color coding for SPK-L as statistics was low, and keep the colour coding for Het-A1 which has more data points included. Darker colours relate to colder temperatures as explained in the legend.

- x-axis ticks and labels are inconsistent between panels a and b → Done

- Y-axis label is inconsistent between panels a and b – unit: kg fuel vs. kg-fuel → Done

Suggestions for simplification: - Use one unique shape for all the ECLIF 1-3 data and differentiate Jet A-1 and SAF / or SAF blends with a different fill color - Use different shapes for SPK-H, SPK-L and Jet A-1 but use the same color map (temperature) for all these points

→ For reasons of clarity we want to stick to the color and symbol style, we have modified the legend and included fuel types to increase clarity.

P1 L26 – "current lean-burn" – an important addition as maintaining low nvPM emissions is crucial; the issue of high ice crystal numbers is linked to fuel sulfur and lubrication oil venting; The latter might require a significant system redesign across all future engine types (a closed loop system).

P8, L6 "volatile organic species form venting of the..." change form to from → Done

P8, L10 "Elorg of 5 mg/kg fuel" – while the authors explain the reasoning behind the assumed oil emission index (manufacturer data), the origin of the El_{org} value is unclear. Further in Methods on Page 36, the authors write that the predicted HC emissions at cruise are 50-66 mg/kg. Was the value of 5 mg/kg assumed as the "activated fraction" to tweak the model to get good agreement with observations? → We thank the reviewer for this comment and clarified accordingly. Only a small fraction of the HC emissions at cruise is activated to nucleate new particles, the rest condenses onto existing particles. We estimate that about 10% of the 50 to 66 mg/kg fuel El_{org} provided by the manufacturer for this engine may contribute to new particle formation, yielding ~5 mg/kg fuel El_{org}. This fraction is supported by observations of volatile particles behind the left and the right engine in the nearfield, as explained in the methods and Fig.4.

P11, L28 "...updates are probably worth considering." – FSC reduction in fossil jet fuel has been the topic of active research and regulatory development. Indeed, especially for lean-burn engines, low-sulfur fuel appears to be key to reducing contrail RF. → Agreed

Table 1 If possible, for quality checks, please add density (at 15 degC) and lower heating value (LHV) with the methods used to the table, if this data is available. → As this information is not required for the content of the manuscript, we have chosen not to include it, in order to maintain focus on the data and information relevant to this study.

El_{nv} for the forced reach burn: median is higher than the reported measurement range – maybe there is a typo? This was carried over from the original version. → In Table 1 we give median and range of El_{nv} for one flight, shown in Fig 1. In Extended Data Table 2 and 3 we give mean El_{nv} and standard deviation for all data points and flights in Fig 2, 3, and 4. We clarify this in the text.

Referee #5 (Remarks to the Author):

The authors present measurements behind the latest generation of aircraft engines burning commonly used and future fuel types. These measurements are absolutely what is needed for a clearer picture for the future of aviation engine design and fuel optimization. They find that even in fuels and engine burn combinations that produce little soot, the ice crystal concentrations produced by the aircraft are similar as to when soot emissions are not actively reduced. This points to the formation of the crystals on other byproducts from the engine combustion e.g. organic vapors and sulfates and from the engine itself, via lubrication oil. These findings confirm previously hypothesized and simulated contrail ice crystal formation processes and as such, are an important addition to our understanding of contrail cirrus formation. That said, the authors often overstate the novelty of their findings rather than presenting the work as a confirmation of what has already been presented from theory and simulations. Nevertheless, due to the broad interest in the climate impacts from aviation and the fact that these measurements clearly show that switching to leaner combustion and newer fuels is not the silver bullet needed to solve the contrail cirrus problem, the manuscript is well suited for publication after some of the novelty is toned down and the following comments are considered.

→ We thank the reviewer for the constructive comments and for recognizing the importance of our in-flight measurements. We will moderate the emphasis on novelty where appropriate.

General comments: Page 1 Line 24-26: This is not exactly true as several of the cited studies have already tested and simulated the ability of volatile particles to initiate ice crystal formation. This should be toned down/reformulated as it is the first time this has been observed behind an aircraft.

→ We have revised the abstract to include relevant theoretical and laboratory studies. While Ponsonby et al. (2024) demonstrated in laboratory experiments that lubrication-oil particles of approximately 100 nm diameter can nucleate ice crystals, particles formed from lubrication-oil vapors at the hot engine exit under cruise conditions are expected to be only a few nanometers in size. The ice-nucleating ability of such small particles can be different due to the curvature effect and to our knowledge has not yet been measured, neither in the lab nor under real aircraft conditions. Determining the corresponding nucleation rates would therefore be highly valuable for improving model representations.

Page 6 Line 18-20: This is higher than the E_{Inv} in the rich burn mode. How was E_{Ice} in those conditions and does this mean that the signal from volatile particles is already present even in the rich burn environment since the E_{Ice} in lean burn is higher than E_{Inv} in rich burn?

→ Yes, this could be one possible explanation. The additional activation of volatile particles in the high-soot regime, besides soot particles, has been reported, for example, by Dischl et al. (2025). Furthermore, our data show near-field non-volatile particle number emissions ($d > 14$ nm) compared to far-field contrail ice measurements under similar engine conditions. Complex plume processing may promote particle growth into larger sizes from near to far field, and emissions of particles < 14 nm, undetected in our near-field data, could also contribute to contrail ice nucleation, potentially explaining the observed differences in emission indices.

Page 6 line 28-30: What else could be responsible for the emissions? I don't really understand the question here as there is clearly a source of particles that is missing and needed to explain the high concentration of ice crystals observed. → Thank you for this comment, our sentence could be misunderstood and we now rephrased it to make our point clearer.

Page 7 line 11-15: It is not immediately clear what this is effectively saying, is this supposed to be motivation for other effects as with these two examples, the difference in E_{Ice} can almost be

explained by the difference in sulfur contents (ratio of 3.2 and 2.6, respectively)? → We thank the reviewer for this comment and rephrased this section to point more clearly to a potential sulfur effect.

Page 8 line 17-20: The word reveal here is a bit misleading. It has already been anticipated that in the absence of soot particles, oil and volatile vapors would act as sources of particles for ice crystals to nucleate on (e.g. Ponsonby et al, 2024; Yu et al, 2024). I would instead use something like “confirm.”

→ Thanks, has been changed

Page 8 lines 25-28: It is not clear what the reasoning behind the smaller crystals is here? Is it that in the absence of larger soot particles, the SS is able to reach higher levels and thus more of the smaller and homogeneously sized sulfate and volatile particles activate into droplets and ice crystals at the same time?

→ Yes, this is what we want to express and we rephrased the sentence accordingly.

Page 9 lines 17-20: It should be made clearer if this related to the soot out competing the oil/organics/sulfates for the available vapor or something else going on. Either way, this feature has already been predicted in previous studies (e.g. Yu et al, 2024), so it is not necessarily a new finding as semi presented here. Instead, it would be better to rephrase this as something that seems more likely from this study and a region that should be sampled. However, based on the results presented here, it looks like the best would be to fly at warmer temperatures and with low sulfur fuel and lean engine combustion.

→ We now clarified that a minimum in contrail ice crystal numbers is simulated between 10^{13} and 3×10^{14} EI_{nv} , extending to lower values than those reported in previous studies (i.e., Kärcher, 2018; Yu et al., 2024).

Page 9 line 26-29: This is an overstatement and not a novel finding. It supports an already anticipated result and as such should be rephrased accordingly. → We take out novel and rephrase accordingly.

The figure captions could be shortened by including some of the information in the text instead.

→ We shortened the figure captions and separate Fig 2 and b into Fig2 and Fig 3.

Minor comments:

At the beginning of the text there are a few acronyms that could be removed as they are only used once or twice and often make the reading a bit distracting. Some examples are:

Page 2 Line 16: ETS acronym is not used again → removed

Page 2 Line 17: CORSIA → removed

Page 4, Line 21: FADEC is only used twice and therefore could be removed and simplified. That said, the details on the engine injectors seem very specific and not of general importance to the study/broad readership, especially without a diagram explaining what is meant by these different locations. Consider shortening the discussion here. → we like to keep this information as it is important for future engine design.

Page 3, Line 10: Consider adding fuel after Jet A1. → done

Page 4 line 26: Is the conventional petroleum-based fossil kerosene mentioned Jet A1? If so, include that here. → done

Page 4 and 5: It might be helpful for the reader to have it clear that 7 fuels were tested (as shown in Table 1) and have them described a bit more clearly in the text. For example the description of SPK-L and H could be streamlined. → We now refer to Table 1 with the fuel compositions.

Page 5 Line 26: consider just writing Jet A1 since it has already been made clear that it is fossil-petroleum based. → changed

Page 6 Line 21-22: Stating that these are the first measurements behind a lean engine seems unnecessary here as this has already been established. → removed

Page 8 Line 22-23: Consider adding a few references here as these findings are very consistent with what is expected of contrail cirrus formation → References have been added.

Page 9 line 13: Add citation to Ponsonby et al, 2014 who showed that above water saturation and at temperatures below the homogeneous freezing temperature, lubricating oils can act as ice-nucleating particles. → Reference to Ponsonby et al., 2024, has been added.

Figure 2: Consider making the outer line of the symbols the temperature. It is not very evident what the temperature of the measurements is with the current colorfill scheme. → We tried several presentation pathways and have selected the clearest.

Page 10 line 23: Can add Ponsonby et al, 2024 here as they did show that lubrication oils act as sites for ice nucleation in the lab. -> has been added. Ponsonby showed the ice nucleation potential for 100 nm sized oil droplets. At cruise, we expect oil particle nucleation in the nanometer particle size range, the ice nucleation rates on those tiny particles still needs to be investigated to inform modelling.

Page 10 line 24-25: Again, Ponsonby et al, 2024 did show this in the lab, it is not clear why the reference is coming in the middle of the sentence where it makes it seem as if the ice nucleation aspect of this statement has not been shown? -> We added Ponsonby et al twice at an earlier place, e.g. in the abstract.

Editorial comments:

Page 2 Line 25: see also Voigt (2025) may read better if it is entirely in parentheses. will be a number

Page 6 line 1: consider adding "like" or something similar instead of "of" → ok

Page 7 line 3: A -> At → changed

Table 1 lines 8-9: Make it clearer that the numbers in the parentheses represent %v if that's true, if not update either way. → changed